# DisPOSE: Projected Polystochastic Diffusion
# for Self-Supervised Multi-View 3D Human Pose Estimation

**Tony Danjun Wang** [1 2]   **Tolga Birdal** [3]   **Nassir Navab** [1 2]   **Lennart Bastian** [1 2 3]

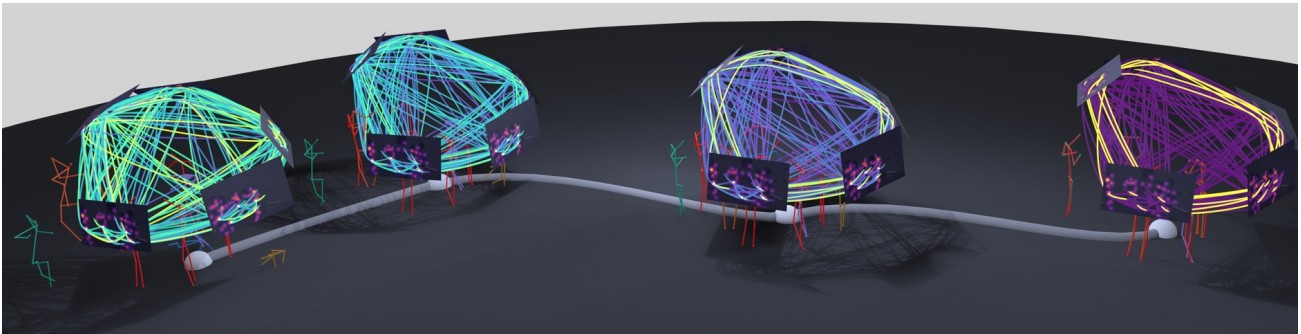

*Figure 1.* **We present DISPOSE**, a novel pose estimation framework that models the discrete problem of associating individuals from multiple camera views as a generative process. By diffusing over the space of polystochastic tensors, DISPOSE learns to recover accurate 3D human associations without requiring 3D ground-truth supervision. As visualized in the trajectory (left to right), the diffusion process progressively resolves ambiguity, evolving from noise into sharp, consistent multi-view associations.

## Abstract

Recovering 3D human poses for multiple individuals from different camera views is a fundamental bottleneck for analyzing interacting behaviors. Existing self-supervised approaches leverage synthetic catalogues of 3D poses; however, this leads to poor generalization in real-world scenarios due to distribution shifts. We therefore introduce DisPOSE, a self-supervised framework that approximates the inherently discrete multi-view person-assignment problem as a generative diffusion process over the space of polystochastic tensors. By employing differentiable Sinkhorn projections during denoising, our model learns to guide solutions toward valid and feasible assignments based on 2D image priors. The complete 3D skeletons of localized individuals are then regressed using a Hypergraph-Convolutional Decoder that explicitly models relational structures and articulated joints across multiple views.

[1]School of Computation, Information, and Technology, Technical University of Munich, Germany [2]Munich Center for Machine Learning, Germany [3]Department of Computing, Imperial College London, United Kingdom. Correspondence to: Tony Danjun Wang <tony.wang@tum.de>.

*Proceedings of the 43$^{rd}$ International Conference on Machine Learning*, Seoul, South Korea. PMLR 306, 2026. Copyright 2026 by the author(s).

The proposed approach outperforms current state-of-the-art self-supervised methods on standard datasets and demonstrates strong performance on a newly proposed benchmark featuring highly occluded scenes from surgical operating rooms. Our diffusion-based localization demonstrates high label efficiency, retaining 99% of its performance with only 10% of the pseudo-labels. Notably, disentangling the assignment and root regression components while maintaining differentiability makes DisPOSE nearly agnostic to different camera arrangements.

## 1. Introduction

Human activity is inherently social and collective. We co-exist to achieve shared goals and engage in group activities, rarely operating individually. Consequently, the spatial configuration of humans must often be interpreted relative to others. In manufacturing and logistics, analyzing the collective movement of workers enables workflow optimization by identifying inefficient movement patterns and physical bottlenecks (Sun et al., 2020). In healthcare, robustly parsing multi-person interactions supports rehabilitation assessment (Avogaro et al., 2023), team coordination (Weiss et al., 2023), and collective behavior during surgical operations (Wang et al., 2025a; Bastian et al., 2023). These applications share a common prerequisite: accurate 3D pose estimation

of multiple individuals that is robust to real-world variations.

Yet, faithfully recovering 3D human poses from visual observations remains profoundly challenging. Monocular approaches, despite benefiting from large-scale annotated datasets, are fundamentally limited by their inability to estimate depth (Zhang et al., 2022) and susceptibility to occlusions (Zheng et al., 2023). These limitations become untenable in safety-critical domains such as collaborative human-robot interaction (Goodrich & Schultz, 2008), and surgical workflow analysis (Garrow et al., 2021), where precision directly impacts clinical outcomes.

Multi-view methods offer a promising alternative by compensating for errors introduced by any single view (Nogueira et al., 2025). Numerous approaches have been proposed (we detail these in Sec. C); however, many methods coalesce around a common recipe: 2D features from each view are aggregated into a 3D voxel grid, from which dense per-voxel likelihoods are regressed, and candidate poses are extracted. This problem naturally bifurcates into two inherently separate problems: *where are the 3D joints* (**regression**) and *to whom do they belong* (**assignment**). Reigning paradigms approximate these two subproblems simultaneously using neural networks, **neglecting the inherent structure of the multi-view assignment problem**.

While optimization-based approaches for solving the assignment component exist, they are either non-differentiable (Wang et al., 2026) or require iterative solvers (Dong et al., 2019) and are thus not easily integrated into learned pipelines. Existing methods are thus either fully supervised or trained on simulated 3D poses that are projected into camera image planes (Srivastav et al., 2024), which introduces domain bias (see Sec. 4.3). Moreover, learning within a fixed voxel grid anchors the representation to specific camera configurations, introducing systematic biases that lead to poor generalization to different viewpoints (see Sec. 4.2).

**Contributions**. To overcome these deficits, we contribute:

- **DISPOSE**: A novel self-supervised framework that effectively *disposes* of the need for explicit 3D supervision. We achieve this by modeling the multi-view assignment problem as a projected diffusion process on polystochastic tensors, coupled with a hypergraph decoder to regress consistent 3D skeletons from the resolved assignments.
- **MM-OR POSE**: A challenging new benchmark for multi-view 3D pose estimation, capturing the highly complex environments of real-world surgical operating rooms
- **Empirically**, DISPOSE establishes a new self-supervised state-of-the-art, improving $AP_{25}$ by 19% on CMU Panoptic and generalizing robustly to novel camera setups (75% mAP vs. baseline 59%). Moreover, our explicit structural modeling yields exceptional data efficiency, retaining 99% of its performance using only 10% of the training data.

## 2. Preliminaries

**Input Representation**. We process the input RGB images with a fine-tuned CNN backbone (i.e., ResNet-50) to infer per-joint heatmaps $\mathbf{H}_v = \{h_{v,j}\}_{j=1}^J \in \mathbb{R}^{J \times H \times W}$ for each view $v$. With $\mathbf{H}_v$, we then obtain 2D root (i.e., hip-joint) candidates $\mathcal{R}_v$ via soft-argmax (Iskakov et al., 2019), which subsequently serve as the nodes for our hypergraph (see Definition 2.2).

**Multi-view Pose Estimation as Assignment**. We consider the task of estimating 3D poses for multiple individuals from a calibrated multi-view setup. Let $V$ denote the number of camera views. Our goal is to recover the 3D poses $\mathcal{P} = \{P_k\}_{k=1}^K$ for all $K$ individuals in the scene, where each pose $P_k \in \mathbb{R}^{J \times 3}$ consists of $J$ Cartesian joint coordinates.

Associating individuals from pairs of views $(i, j)$ can be phrased as a linear assignment problem in the most direct sense (Munkres, 1957). MVPose (Dong et al., 2019) models this association by relaxing the assumption of a binary-valued assignment matrix $A_{ij}$ and considering doubly-stochastic matrices through marginalization:

**Definition 2.1** (Mode-$v$ Marginal). For a nonnegative tensor $\mathcal{X} \in \mathbb{R}_+^{N \times \cdots \times N}$ of order $V$, the *mode-$v$ marginal* $\mathcal{M}_v(\mathcal{X})$ is the vector obtained by summing over all modes except $v$:

$$\left(\mathcal{M}_v(\mathcal{X})\right)_{i_v} = \sum_{\{i_u : u \neq v\}} \mathcal{X}_{i_1, \ldots, i_V}. \tag{1}$$

When $V = 2$, we recover doubly-stochastic matrices widely adopted in matching problems (Sarlin et al., 2020), and span the *Birkhoff polytope* (Birdal & Simsekli, 2019; Xie et al., 2025), whose vertices are the set of permutation matrices.

A *cycle-consistent* matching **across all cameras** can then be enforced through rank minimization, which would otherwise be challenging to formulate in combinatorial terms (Dong et al., 2019). The resulting formuation is a constrained convex optimization problem and can be solved, e.g., via ADAM (Neal et al., 2011).

Our formulation adopts a recent rephrasing of this problem as an association over all cameras jointly (without post-hoc constraints), finding an optimal subset of hyperedges covering each detection exactly once (Wang et al., 2026):

**Definition 2.2** (Multi-view Correspondence Hypergraph). Let $\mathcal{R}_v = \{r_{v,i}\}_{i=1}^{N_v}$ denote the set of $N_v$ candidate 2D root detections in view $v$. A *multi-view correspondence hypergraph* is defined as $\mathcal{G} = (\mathcal{D}, \mathcal{E})$, where:

- $\mathcal{D} = \bigcup_{v=1}^V \mathcal{R}_v$ is the node set comprising all detections across views, and
- each hyperedge $e \in \mathcal{E}$ is a tuple $(r_1^{i_1}, \ldots, r_V^{i_V})$ containing at most one detection per view, representing a candidate 3D person.

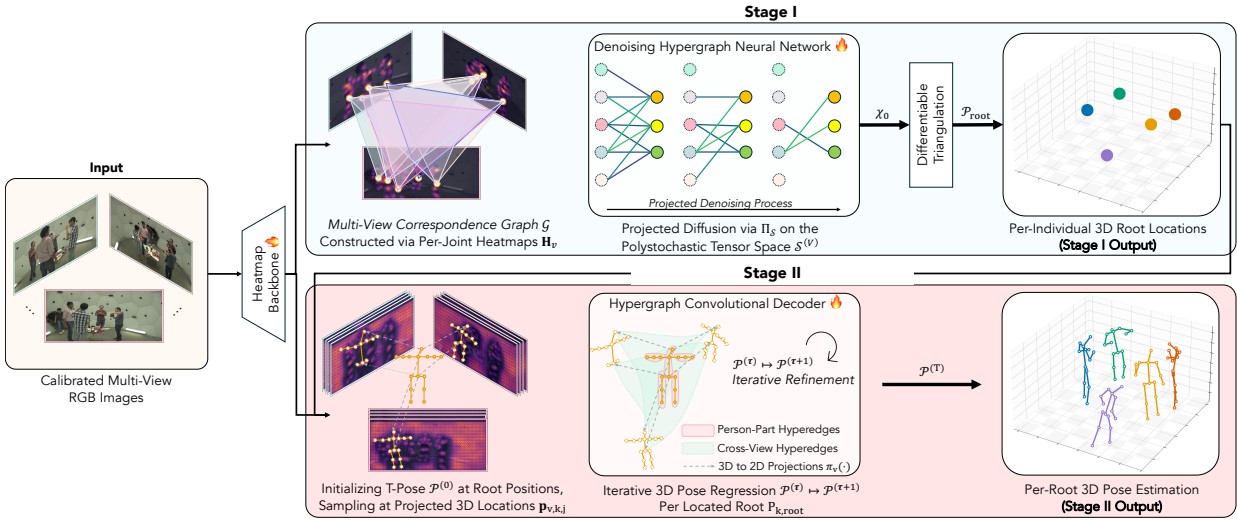

*Figure 2.* **Overview of the DɪsPOSE framework. Stage I (Root Regression)** constructs the multi-view correspondence hypergraph $\mathcal{G}$ and solves higher-order correspondences via projected diffusion over the polystochastic set $\mathcal{S}^{(V)}$. The Sinkhorn projection $\Pi_{\mathcal{S}}$ enforces marginal feasibility, yielding polystochastic tensors $\mathcal{X}$ and 3D roots $\mathcal{P}_{\text{root}}$. **Stage II (Pose Regression)** initializes a canonical template $\mathcal{P}^{(0)}$ at the triangulated root positions and samples intermediate backbone features. A hypergraph convolutional decoder iteratively refines $\mathcal{P}^{(t)} \mapsto \mathcal{P}^{(t+1)}$ to obtain full 3D poses $\mathcal{P}^{(T)}$ by aggregating cross-view evidence and modeling anatomical person-part structures.

In practice, we follow (Wang et al., 2026) and first prune geometrically implausible candidate hyperedges, retaining only a small support set $\tilde{\mathcal{E}} \subset \mathcal{E}$. All subsequent diffusion and projection steps operate on this retained support rather than the full dense $N^V$ tensor. An optimal set of hyperedges partitioning individuals from all camera views can then be recovered via an iterative discrete solver (Wang et al., 2026).

**Permutation and Polystochastic Tensors**. In contrast to pairwise assignment formulations (Dong et al., 2019), which rely on bi-stochastic matrices, we represent this global assignment as an order-$V$ binary tensor $\mathcal{X}$; the adjacency tensor of the hypergraph $\mathcal{G}$[1]. To ensure physical validity (a node cannot belong to multiple people), $\mathcal{X}$ must belong to the set of *permutation tensors* $\mathcal{A}^{(V)}$ (Alexeev et al., 2011):

**Definition 2.3** (Permutation and Polystochastic Tensors). The set of *permutation tensors* is defined as:

$$\mathcal{A}^{(V)} := \left\{ \mathcal{X} \in \{0,1\}^{N \times \cdots \times N} : \mathcal{M}_v(\mathcal{X}) = \mathbf{1}, \ \forall v \right\}. \quad (2)$$

The set of *polystochastic tensors* is a continuous relaxation:

$$\mathcal{S}^{(V)} := \left\{ \mathcal{X} \in \mathbb{R}_+^{N \times \cdots \times N} : \mathcal{M}_v(\mathcal{X}) = \mathbf{1}, \ \forall v \right\}. \quad (3)$$

**Remark 2.4** (Geometric Optimality). While the permutation constraints guarantee physical validity, they do not ensure geometric consistency. We define the *optimal* assignment $\mathcal{X}^* \in \mathcal{A}^{(V)}$ as the unique tensor that associates 2D

---

[1]Perfect matching is typically violated by occlusions or limited overlap between cameras; we therefore augment each view with a "dustbin", allowing it to absorb unmatched identities. The hypergraph is therefore $V$-uniform, with an adjacency tensor.

detections corresponding to the same 3D identity, thereby satisfying the global geometric constraints.

As the set $\mathcal{A}^{(V)}$ is discrete, we formulate our learning problem over the relaxed set $\mathcal{S}^{(V)}$ (Chang et al., 2016).

**Multi-Marginal Sinkhorn Normalization**. Projecting unconstrained scores onto the polystochastic manifold $\mathcal{S}^{(V)}$ is possible through *multi-marginal Sinkhorn normalization* (Lin et al., 2022; Friedland, 2020), the higher-dimensional analog of the famous Sinkhorn-Knopp algorithm (Sinkhorn & Knopp, 1967). Using the notation of (Mena et al., 2018), we define:

**Definition 2.5** (Mode-$v$ Normalization). The *mode-$v$ normalization operator* $\mathcal{T}_v$ normalizes a nonnegative tensor $\mathcal{X}$ to have unitary sum along mode $v$:

$$\mathcal{T}_v(\mathcal{X}) = \mathcal{X} \odot \frac{\mathbf{1}}{\mathcal{M}_v(\mathcal{X})}, \quad (4)$$

where the ratio is broadcast along all modes except $v$.

**Definition 2.6** (Multi-Marginal Sinkhorn). Let $X$ be a tensor of unconstrained scores. The *Sinkhorn operator* $S(X)$ is defined by cyclically applying $\mathcal{T}_v$ for all views $v = 1, \dots, V$:

$$S^0(X) = \exp(X), \quad (5)$$

$$S^\ell(X) = \mathcal{T}_V\left(\mathcal{T}_{V-1}\left(\cdots \mathcal{T}_1\left(S^{\ell-1}(X)\right)\cdots\right)\right), \quad (6)$$

$$S(X) = \lim_{\ell \to \infty} S^\ell(X). \quad (7)$$

The iterates $S^\ell(X)$ converge to a tensor $S(X) \in \mathcal{S}^{(V)}$ whose mode-$v$ marginals all equal $\mathbf{1}$ (Carlier, 2022; Piran

et al., 2024). In practice, we use the truncated operator $\Pi_{\mathcal{S}}(X) = S^L(X)$ with a fixed number of iterations $L$. To avoid materializing the dense $N^V$ tensor, we implement this projection on the retained hyperedge support $\tilde{\mathcal{E}}$. Our projected diffusion model, presented next, leverages this formulation to approximate solutions to the assignment problem on the hypergraph in Definition 2.2 (Wang et al., 2026).

## 3. DISPOSE

We propose a two-stage framework to recover multi-person 3D poses. We stratify the task into: (1) **Root Regression**, where we solve the higher-order association problem to triangulate 3D roots (i.e., hip-joints), and (2) **Pose Regression**, where we regress full-body poses conditioned on these roots (see Fig. 2).

### 3.1. Projected Polystochastic Diffusion (Stage I)

We frame 2D root association as a generative denoising problem over the assignment tensor $\mathcal{X}$. Our core insight is to diffuse over the space of valid assignments.

**Diffusion on Polystochastic Tensors.** Standard Gaussian diffusion operates in unconstrained Euclidean space, whereas valid multi-view assignments must satisfy the marginal constraints of Definition 2.3. We thus parameterize assignments via their log-potentials $\phi(\mathcal{X}) = \log(\mathcal{X})$, as performing Gaussian diffusion in this log-score space has been shown to be more numerically stable (Schmitzer, 2019). It is a natural parameterization, as the Sinkhorn algorithm solves entropy-regularized optimal transport, whose solutions are exponentials of unconstrained dual variables in $\mathbb{R}$ (Cuturi, 2013; Mena et al., 2018; Peyré & Cuturi, 2019). Feasibility can then be enforced by projecting back onto $\mathcal{S}^{(V)}$ via the Sinkhorn operator $\Pi_{\mathcal{S}}$, following the projected diffusion framework (Lou & Ermon, 2023; Fishman et al., 2024).

**Definition 3.1** (Projected Forward Diffusion). Given a ground truth assignment $\mathcal{X}_0 \in \mathcal{A}^{(V)}$ with latent coordinates $\mathbf{u}_0 = \phi(\mathcal{X}_0)$, the forward process applies Gaussian corruption in log-score space:

$$\mathbf{u}_t = \sqrt{\bar{\alpha}_t}\,\mathbf{u}_0 + \sqrt{1 - \bar{\alpha}_t}\,\boldsymbol{\epsilon}, \quad \boldsymbol{\epsilon} \sim \mathcal{N}(0, \mathbf{I}), \quad (8)$$

followed by projection onto the feasible set: $\mathcal{X}_t = \Pi_{\mathcal{S}}(\mathbf{u}_t)$.

This ensures that $\mathcal{X}_t$ remains a valid polystochastic tensor throughout the process, providing meaningful structural cues to the denoiser.

**Definition 3.2** (Projected Reverse Diffusion). Given a denoiser $f_\theta(\mathcal{X}_t, t)$ predicting clean scores $\hat{\mathbf{u}}_0$, the reverse process computes the implied noise estimate

$$\hat{\boldsymbol{\epsilon}}_t = \frac{\mathbf{u}_t - \sqrt{\bar{\alpha}_t}\,\hat{\mathbf{u}}_0}{\sqrt{1 - \bar{\alpha}_t}}, \quad (9)$$

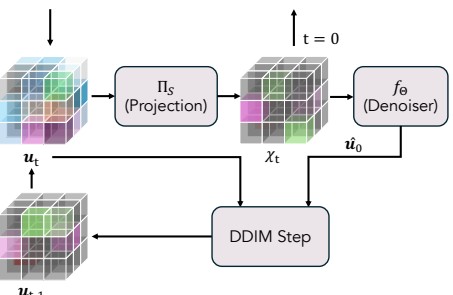

*Figure 3.* **Projected Reverse-Time Generation.** We depict a single step of the denoising process for root regression. We start by projecting noisy latent scores $\mathbf{u}_t$ onto the polystochastic manifold via the Sinkhorn operator $\Pi_{\mathcal{S}}$, obtaining a valid assignment tensor $\mathcal{X}_t$. Conditioned on this feasible state, the hypergraph denoiser $f_\theta$ predicts clean scores $\hat{\mathbf{u}}_0$, after which a DDIM update yields $\mathbf{u}_{t-1}$. Finally, at $t = 0$, we take $\mathcal{X}_0$ as the valid assignment.

performs a DDIM update in score space,

$$\mathbf{u}_{t-1} = \sqrt{\bar{\alpha}_{t-1}}\,\hat{\mathbf{u}}_0 + \sqrt{1 - \bar{\alpha}_{t-1} - \sigma_t^2}\,\hat{\boldsymbol{\epsilon}}_t + \sigma_t \mathbf{z}, \quad (10)$$

where $\mathbf{z} \sim \mathcal{N}(0, \mathbf{I})$ and $\sigma_t$ controls sampling stochasticity, then projects the results back onto the feasible set:

$$\mathcal{X}_{t-1} = \Pi_{\mathcal{S}}(\mathbf{u}_{t-1}) \in \mathcal{S}^{(V)}. \quad (11)$$

In Fig. 3, we illustrate one reverse-time step of our projected diffusion. We train a hypergraph neural network (Chien et al., 2022) as denoiser $f_\theta(\mathcal{X}_t, t)$ to predict the clean scores:

$$\hat{\mathbf{u}}_0 = f_\theta(\mathcal{X}_t, t). \quad (12)$$

By enforcing $\Pi_{\mathcal{S}}$ at every step, we guide the generation process along the polystochastic manifold. We provide the full algorithm as pseudo-code in the appendix (alg. 1).

**Greedy Rounding and Triangulation.** Upon convergence, we recover the soft assignment $\hat{\mathcal{X}}_0 = \Pi_{\mathcal{S}}(\hat{\mathbf{u}}_0)$. We then extract discrete matches via *greedy rounding*, where we iteratively select the hyperedge with maximum probability in $\hat{\mathcal{X}}_0$ and suppress collinear entries. We finally triangulate the disjoint hyperedges to initialize the pose regression in Stage II.

### 3.2. Pose Regression via Graph Refinement (Stage II)

Given the 3D triangulated root locations, we estimate full-body poses $\mathcal{P} = \{P_k\}_{k=1}^K$ via iterative refinement (Liao et al., 2024). We formulate pose regression as a sequence of hypergraph operators (Bai et al., 2021) that aggregate information across the relational structures induced by visible poses in all views.

Let $\mathcal{P}^{(\tau)} = \{P_k^{(\tau)}\}_{k=1}^K$ denote the set of pose hypotheses at refinement step $\tau$, where each pose $P_k^{(\tau)} \in \mathbb{R}^{J \times 3}$ comprises joint coordinates $\{P_{k,j}^{(\tau)}\}_{j=1}^J$. For each person $k$, we

initialize $P_k^{(0)}$ using a canonical T-pose (Liao et al., 2024) anchored at the estimated 3D root location. We then apply a decoder with T refinement layers. Each layer maps $\mathcal{P}^{(\tau)} \mapsto \mathcal{P}^{(\tau+1)}$ by *(i)* sampling multi-view image features at the current 2D projections and predicting per-joint 2D corrections, *(ii)* triangulating updated 3D joints, and *(iii)* enforcing intra-person anatomical consistency to form features for the next refinement step.

**Multi-View Hypergraph For 2D Refinement**. At layer $\tau$, we project each current 3D joint hypothesis into view $v$:

$$\mathbf{p}_{v,k,j}^{(\tau)} = \pi_v\left(P_{k,j}^{(\tau)}\right) \in \mathbb{R}^2, \qquad (13)$$

and sample view-specific image features around $\mathbf{p}_{v,k,j}^{(\tau)}$ via projective attention (Xia et al., 2022). We treat each projected joint $(v, k, j)$ as a node and construct a multi-view hypergraph whose edges capture two complementary groupings: *(i)* intra-view person context (all joints of person $k$ within view $v$), and *(ii)* inter-view joint-type context (the same joint $j$ of person $k$ across views). With hypergraph convolutions (Bai et al., 2021), we aggregate evidence within each edge, yielding refined per-node features which predict a 2D offset $\Delta\mathbf{p}_{v,k,j}^{(\tau)}$ and a per-node confidence $c_{v,k,j}^{(\tau)}$.

**Geometry Update Via Triangulation**. We update 3D joints by triangulating from the corrected multi-view projections,

$$P_{k,j}^{(\tau+1)} = \text{Triangulate}\Big( \left\{\mathbf{p}_{v,k,j}^{(\tau)} + \Delta\mathbf{p}_{v,k,j}^{(\tau)}\right\}_{v=1}^V ; \\ \left\{c_{v,k,j}^{(\tau)}\right\}_{v=1}^V \Big), \qquad (14)$$

where $c_{v,k,j}^{(\tau)}$ weights the contribution of each view. We solve this weighted algebraic triangulation differentiably by SVD (Iskakov et al., 2019). This provides an explicit multi-view geometric consistency step within each refinement layer.

**Person-Part Hypergraph For Anatomical Consistency**. To enforce anatomical coherence, we apply a second, person-part hypergraph over the joints of each triangulated person $k$. The edges connect *(i)* all joints of a person and *(ii)* joints within coarse body parts (e.g., shoulder, elbow, wrist), enabling message passing that captures skeletal dependencies and suppresses locally ambiguous updates. The resulting anatomy-aware features are carried into the next layer, refining subsequent multi-view correction and triangulation.

### 3.3. Supervision Strategy

We train our model under *weak supervision* by leveraging off-the-shelf 2D pose detections (Xu et al., 2022) and correspondence 3D pseudo-labels (Wang et al., 2026), without requiring any ground-truth 3D annotations. Our final training objective combines correspondence learning and multi-stage pose regression:

$$\mathcal{L} = \lambda_{\text{diff}}\,\mathcal{L}_{\text{diff}} + \sum_{\tau=1}^{\text{T}} \left( \lambda_{\text{coord}}\,\mathcal{L}_{\text{coord}}^{(\tau)} + \lambda_{\text{hm}}\,\mathcal{L}_{\text{hm}}^{(\tau)} + \lambda_{\text{geo}}\,\mathcal{L}_{\text{geo}}^{(\tau)} \right).$$

We detail the individual components below.

**Assignment Loss ($\mathcal{L}_{\text{diff}}$)**. The diffusion-based assignment model is supervised using the pseudo ground-truth assignment $\mathcal{X}^\star \in \mathcal{A}^{(V)}$. Let $\mathbf{u}^\star = \phi(\mathcal{X}^\star)$ be its latent coordinate representation, then we define the latent score matching loss:

$$\mathcal{L}_{\text{diff}} = \mathbb{E}_t\left[\left\|\hat{\mathbf{u}}_0 - \mathbf{u}^\star\right\|_2^2\right]. \qquad (15)$$

**Pose Regression Losses**. We apply deep supervision to the pose regression decoder at each layer $\tau$. The primary supervision signal is provided by 2D human poses obtained from an off-the-shelf 2D pose detector. For each view $v$, we project the predicted 3D joints and penalize disagreement with detected keypoints using a confidence-weighted loss,

$$\mathcal{L}_{\text{coord}}^{(\tau)} = \sum_{v,k,j} s_{v,k,j}\,\rho\Big(\pi_v\left(P_{k,j}^{(\tau)}\right) - \hat{\mathbf{q}}_{v,k,j}\Big), \qquad (16)$$

where $\hat{\mathbf{q}}_{v,k,j}$ and $s_{v,k,j}$ are the 2D detector keypoints and confidences, and $\rho(\cdot)$ is a robust loss (e.g., $\ell_1$). To improve robustness to occlusion, we include $\mathcal{L}_{\text{hm}}^{(\tau)}$, an *asymmetric* $\ell_2$ heatmap consistency loss penalizing missing detector-supported joints without penalizing plausible predictions where the detector is uncertain. Finally, $\mathcal{L}_{\text{geo}}^{(\tau)}$ imposes geometric regularization to anchor predictions to the global consensus, enforce affine invariance, and ensure valid multi-view triangulation (see Sec. A.2 for more details).

### 3.4. Implementation Details

**Training Setup**. We implement DisPOSE in PyTorch and train on a single NVIDIA A40 GPU via Adam with a learning rate of $4e^{-4}$ and a batch size of 2 for 50,000 iterations. For the diffusion process, we use 1000 timesteps during training and sample using 10 DDIM steps. The Sinkhorn projection $\Pi_{\mathcal{S}}$ uses $L = 4$ iterations. We use the same data augmentation strategy as SelfPose3D. In the appendix, we provide details regarding network specifications (Sec. A.4) and data augmentation (Sec. A.5).

**Diffusion Network**. Our denoiser $f_\theta$ is a conditioned hypergraph message passing network (Chien et al., 2022) that predicts clean latent scores $\hat{\mathbf{u}}_0$ from the feasible diffusion state $\mathcal{X}_t$. We condition the denoising dynamics on hyperedge cues $z_e \in [0, 1]$. These cues represent the geometric consistency costs defined in our *higher-order assignment formulation* (see Sec. 2) and effectively guide the diffusion process using the same geometric priors as in optimization-based approaches (Wang et al., 2026).

*Table 1.* Quantitative comparison for **pose estimation (stage I and II)** on the CMU Panoptic dataset. [†] uses 9 temporal frames for input. Best self-supervised results are **highlighted**.

| Method | AP (mm) ($\uparrow$) | | | | Recall ($\uparrow$) | MPJPE |
|---|---|---|---|---|---|---|
| | 25 | 50 | 100 | 150 | @500 | ($\downarrow$) |
| *Fully-Supervised* | | | | | | |
| VoxelPose (Tu et al., 2020) | 83.59 | 98.33 | 99.76 | 99.91 | – | 17.68 |
| Plane Sweep (Lin & Lee, 2021) | 92.12 | 98.96 | 99.81 | 99.84 | – | 16.75 |
| MvP (Wang et al., 2021) | 92.28 | 96.60 | 97.45 | 97.69 | – | 15.76 |
| (Wu et al., 2021) | 93.93 | 98.93 | 99.78 | 99.90 | 99.97 | 15.63 |
| Faster Voxel (Ye et al., 2022) | 85.22 | 98.08 | 99.32 | 99.48 | – | 18.26 |
| TEMPO (Choudhury et al., 2023) | 89.01 | 99.08 | 99.76 | 99.93 | – | 14.68 |
| MVGFormer (Liao et al., 2024) | 92.32 | 97.93 | 99.32 | 99.55 | 99.86 | 15.99 |
| Voxel.+3DSA (Chen & Tsai, 2025) | 94.20 | 98.49 | 99.21 | 99.31 | – | 13.98 |
| MV-SSM (Chharia et al., 2025) | 93.50 | – | – | – | – | 15.70 |
| *Optimization-Based* | | | | | | |
| ACTOR (Pirinen et al., 2019) | – | – | – | – | – | 168.40 |
| MvPose (Dong et al., 2019) | 37.63 | 95.70 | 97.84 | 98.28 | 99.60 | 26.46 |
| COMPOSE (Wang et al., 2026) | 54.66 | 97.27 | 98.94 | 99.17 | 99.83 | 23.62 |
| *Self-Supervised* | | | | | | |
| SelfPose3d (Srivastav et al., 2024) | 55.13 | 96.44 | 98.46 | 98.98 | 99.60 | 24.47 |
| DSP[†] (Liu & Zhang, 2025) | 57.60 | 86.10 | 94.00 | – | – | 23.10 |
| DisPOSE (Ours) | **68.59** | **98.59** | **99.60** | **99.80** | **99.91** | **21.20** |

**Hypergraph Decoder**. To efficiently process the higher-order dependencies described in Sec. 3.2, we employ the attention-based hypergraph convolution mechanism (Bai et al., 2021). We apply this operator to both the multi-view and person-centric structures to compute importance weights over incident node–hyperedge relations, allowing the model to adaptively aggregate messages from relevant views and body parts without explicit edge weights.

## 4. Experiments and Results

### 4.1. Datasets and Evaluation Metrics

We evaluate our proposed method on three established datasets: **CMU Panoptic**, **Shelf**, and **Campus**. Additionally, we propose **MM-OR POSE**, a *new* dataset for multi-view 3D human pose estimation, featuring challenging, visually distinct scenarios from surgical operating rooms, thus representing a significant deviation from existing datasets.

For CMU Panoptic, Shelf, and Campus, we follow the standard train/test protocol established in (Tu et al., 2020), using the defined camera setup and train/test splits. We provide further details on the datasets in the appendix (Sec. B.1)

**MM-OR POSE** extends the MM-OR dataset (Özsoy et al., 2025) to facilitate multi-view 3D pose estimation in challenging surgical environments characterized by loose attire, unusual poses, and severe occlusions (Bastian et al., 2023; Wang et al., 2025b). As the original dataset (simulated knee surgeries via 5 cameras) lacks 3D human pose labels, we manually annotate 750 test frames to establish a new benchmark. We release these annotations to the community; see Sec. B.1.1 for details.

**CMU Panoptic** (Joo et al., 2015) is a large-scale dataset capturing diverse social activities in an indoor dome. It

*Table 2.* Quantitative comparison for **pose estimation (stage I & II)** on the proposed MM-OR POSE. Best results are **highlighted**.

| Method | AP (mm) ($\uparrow$) | | | Recall ($\uparrow$) | MPJPE ($\downarrow$) |
|---|---|---|---|---|---|
| | 50 | 100 | 150 | @500 | (mm) |
| *Self-Supervised* | | | | | |
| SelfPose3d (Srivastav et al., 2024) | 23.38 | 69.68 | 82.82 | 94.33 | 70.69 |
| DisPOSE (Ours) | **47.06** | **83.59** | **92.01** | **97.04** | **56.91** |

features moderate occlusion solely from inter-person interactions, as the environment lacks external obstacles.

**Shelf** (Belagiannis et al., 2014) depicts four individuals interacting with a wooden shelf in a confined indoor space, observed by 5 calibrated RGB cameras.

**Campus** (Belagiannis et al., 2014) features three subjects navigating an outdoor courtyard, captured by a minimal setup of 3 calibrated RGB cameras.

**Evaluation Metrics**. Following standard protocols (Tu et al., 2020), we evaluate CMU Panoptic and MM-OR Pose using Average Precision (AP) at various thresholds, Recall at 500 mm, and Mean Per Joint Position Error (MPJPE). We report mAP as the mean AP over thresholds of 25, 50, 75, 100, 125, and 150 mm. For Shelf & Campus, we report the Percentage of Correct Parts (PCP) (Belagiannis et al., 2014).

### 4.2. Quantitative Results

We present quantitative results on CMU Panoptic, MM-OR Pose, as well as Shelf & Campus.

**3D Human Pose Estimation Results**. We compare our method against state-of-the-art fully supervised, optimization-based, and self-supervised multi-view, multi-human 3D pose estimation methods. If not otherwise specified, we use the results reported in the original papers.

Tab. 1 reports the quantitative results on CMU Panoptic. DisPOSE achieves state-of-the-art performance among self-supervised methods, outperforming the previous best method, DSP (Liu & Zhang, 2025), across all metrics. We achieve an MPJPE of 21.20 mm, an improvement of 8% over DSP (23.10 mm) and 13% over SelfPose3D (24.47 mm). Notably, our method shows superior precision at strict thresholds, scoring 68.59 in $AP_{25}$ vs. 57.60 for DSP.

Tab. 2 presents the results on the proposed MM-OR Pose dataset. We do not report $AP_{25}$ as the challenging nature of the dataset makes it profoundly difficult to annotate joints with millimeter precision (see Sec. B.1.1). Notably, compared to CMU Panoptic, all methods experience a significant performance drop due to the challenging nature of the surgical environment, peaking at 92.04% for $AP_{150}$ (vs. 99.80% on CMU Panoptic). However, DisPOSE remains the most robust compared to SelfPose3D, achieving a recall of 97.04% vs. 94.33%, with 10% higher $AP_{150}$.

*Table 3.* Quantitative comparison for **pose estimation (stage I and II)** on Shelf and Campus (PCP %). $^\dagger$ takes 81 temporal frames as input. Best self-supervised results are highlighted in **bold**.

| Method | Shelf (PCP %) (↑) | | | | Campus (PCP %) (↑) | | | |
|---|---|---|---|---|---|---|---|---|
| | Act 1 | Act 2 | Act 3 | Avg. | Act 1 | Act 2 | Act 3 | Avg. |
| *Fully Supervised* | | | | | | | | |
| (Ershadi-Nasab et al., 2018) | 93.3 | 75.9 | 94.8 | 88.0 | 94.2 | 92.9 | 84.6 | 90.6 |
| VoxelPose (Tu et al., 2020) | 99.3 | 94.1 | 97.6 | 97.0 | 97.6 | 93.8 | 98.8 | 96.7 |
| (Wu et al., 2021) | 99.3 | 96.5 | 97.3 | 97.7 | – | – | – | – |
| MvP (Wang et al., 2021) | 99.3 | 95.1 | 97.8 | 97.4 | 98.2 | 94.1 | 97.4 | 96.6 |
| Faster Voxel. (Ye et al., 2022) | 99.4 | 96.0 | 97.5 | 97.6 | 96.5 | 94.1 | 97.9 | 96.2 |
| TEMPO (Choudhury et al., 2023) | 99.3 | 95.1 | 97.8 | 97.4 | 97.7 | 95.5 | 97.9 | 97.3 |
| *Optimization-Based* | | | | | | | | |
| 3DPS (Belagiannis et al., 2014) | 75.3 | 69.7 | 87.6 | 77.5 | 93.5 | 75.7 | 84.4 | 84.5 |
| MvPose (Dong et al., 2019) | 98.8 | 94.1 | 97.8 | 96.9 | 97.6 | 93.3 | 98.0 | 96.3 |
| COMPOSE (Wang et al., 2026) | 99.8 | 92.4 | 96.3 | 96.2 | 99.4 | 94.3 | 98.1 | 97.3 |
| *Self-Supervised* | | | | | | | | |
| SelfPose3d (Srivastav et al., 2024) | 97.2 | 90.3 | 97.9 | 95.1 | 92.5 | 82.2 | 89.2 | 87.9 |
| DSP $^\dagger$ (Liu & Zhang, 2025) | 99.1 | 92.8 | **98.3** | 96.7 | 94.9 | 91.0 | 92.4 | 92.8 |
| DisPOSE (Ours) | **99.5** | **94.1** | 97.8 | **97.1** | **98.8** | **93.7** | **94.3** | **95.6** |

*Table 4.* Ablation study for both **root regression (stage I) and pose regression (stage II)** on differing camera arrangements. We compare SelfPose3d and DisPOSE on varying camera arrangements on CMU Panoptic. Best results are highlighted in **bold.**

| Setup / Method | Root | | | Pose | | |
|---|---|---|---|---|---|---|
| | mAP (%)↑ | Rec. (%)↑ | MDE (mm)↓ | mAP (%)↑ | Rec. (%)↑ | MPJPE (mm)↓ |
| *CMU1* (7 cameras) | | | | | | |
| SelfPose3d (Srivastav et al., 2024) | 50.99 | 98.00 | 74.22 | 74.50 | 97.98 | 42.08 |
| DisPOSE (Ours) | **73.42** | **99.90** | **46.34** | **91.97** | **99.92** | **23.26** |
| *CMU2* (7 cameras) | | | | | | |
| SelfPose3d (Srivastav et al., 2024) | 32.31 | 94.58 | 115.13 | 59.06 | 94.32 | 67.90 |
| DisPOSE (Ours) | **78.94** | **99.52** | **38.28** | **83.01** | **99.80** | **30.44** |
| *CMU3* (4 cameras) | | | | | | |
| SelfPose3d (Srivastav et al., 2024) | 29.94 | 85.86 | 114.91 | 61.43 | 83.96 | 49.61 |
| DisPOSE (Ours) | **69.20** | **90.78** | **43.37** | **75.57** | **90.75** | **35.72** |
| *CMU4* (4 cameras) | | | | | | |
| SelfPose3d (Srivastav et al., 2024) | 31.19 | 97.92 | 132.07 | 62.85 | 98.32 | 72.91 |
| DisPOSE (Ours) | **71.89** | **99.86** | **50.29** | **87.02** | **99.85** | **28.10** |

Tab. 3 compares our method on Shelf and Campus. DisPOSE outperforms the self-supervised baselines on both datasets. On Shelf, we achieve an average PCP of 97.1%, surpassing DSP (96.7%) and SelfPose3d (95.1%). On Campus, we observe a significant margin of improvement, achieving 95.6% PCP compared to 92.8% for DSP and 87.9% for SelfPose3d, demonstrating the robustness of our approach across different environments.

**Generalization Results**. As shown in Tab. 4, DisPOSE generalizes significantly better to unseen camera configurations than the baseline. While SelfPose3d shows a drastic performance drop on setups that differ from the training set, dropping to 32.31% and 29.94% root mAP on CMU2 and CMU3, respectively, our method maintains high precision, consistently achieving over 69% root mAP across all tested arrangements. Notably, our method leverages denser camera arrays more effectively, improving from an average of 70.5% root mAP on 4-camera setups to 76.2% on 7-camera setups. In contrast, SelfPose3d demonstrates limited adaptability to increased view counts; for instance, its performance on the 7-camera CMU2 setup (32.31%) is remarkably similar to the 4-camera CMU4 setup (31.19%). All experiments are performed without fine-tuning; for more details on the camera arrangements, please refer to Sec. B.4 in the appendix.

### 4.3. Qualitative Results

Fig. 4 illustrates a qualitative example from CMU Panoptic, featuring a challenging out-of-distribution scenario in which a toddler plays on the ground. SelfPose3D fails to detect the subject entirely. This likely stems from training on synthetic catalogs of 3D poses, which introduce a simulation bias that prevents generalization to unseen scales (e.g., toddlers) or unusual poses (e.g., crawling on the floor). In contrast, DisPOSE successfully localizes and estimates the toddler's 3D pose. By avoiding restrictive synthetic priors and instead learning structured assignments directly from the multi-view input, our method demonstrates superior robustness to real-world variations in scale and position.

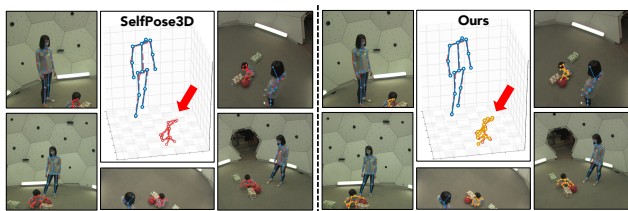

*Figure 4.* **Qualitative Example.** We compare SelfPose3D and DisPOSE (Ours) on the CMU Panoptic dataset. Ground truth is shown in red, predictions in blue and orange.

We present qualitative results of our proposed method against SelfPose3D on the proposed MM-OR Pose dataset in Fig. 5. We show a frame in which two surgeons interact closely, with one bending over the operating table, creating a scenario with severe occlusion and a non-canonical posture. As shown in the figure, SelfPose3D fails to accurately capture the unusual pose (highlighted in red). In contrast, DisPOSE successfully estimates the 3D poses of both surgeons, robustly handling the challenging bent-over posture. However, we note that, despite the overall improvement, localizing specific joints, such as the nose and hip, remains difficult due to the significant visual obstruction caused by surgical helmets and sterile gowns.

### 4.4. Ablation Studies

We conduct extensive ablation studies to analyze the contributions of different components of our proposed method.

**Assignment Module (Stage I).** In Tab. 5, we evaluate the performance of our diffusion-based root regression module on the CMU Panoptic dataset. We refer to Sec. B.3 in the ap-

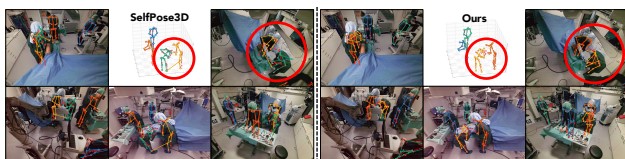

*Figure 5.* **Qualitative Example**. We compare SelfPose3D and DisPOSE (Ours) on the newly proposed MM-OR Pose dataset. Ground truth is shown in red, prediction in green, orange, yellow, and blue. We provide a larger version in the appendix (see Fig. 11)

pendix for experimental details. DisPOSE significantly outperforms the self-supervised SelfPose3D, improving mAP by over 13% (81.49% vs 68.21%) and reducing the mean distance error by 19mm (35.40 mm vs 54.82 mm). Notably, our self-supervised results surpass fully-supervised methods, including VoxelPose and Faster VoxelPose.

Given the stochastic nature of our diffusion-based approach, we report the mean and standard error over 10 runs with different random seeds. As shown in the table, the standard error is negligible (e.g., $\pm 1e\text{-}4$ for mAP); therefore, we omit standard error reporting in the remaining tables.

*Table 5.* Quantitative comparison of our assignment-based **root regression (stage I)** on the CMU Panoptic dataset (Joo et al., 2015). Best self-supervised results are highlighted in **bold**. † uses an additional learned implicit distance function, ‡ uses triangulation from predicted (pairwise) correspondences.

| Method | mAP (%) | Recall (@500mm) | MDE (mm) $\downarrow$ |
|---|---|---|---|
| *Fully-Supervised* | | | |
| VoxelPose (Tu et al., 2020) | 72.93 | 99.80 | 38.47 |
| Faster Voxel. (Ye et al., 2022) | 69.11 | 99.67 | 49.93 |
| (Wu et al., 2021)† | 82.63 | 99.97 | 33.25 |
| (Wu et al., 2021)‡ | 60.75 | 98.81 | 39.58 |
| *Self-Supervised* | | | |
| SelfPose3d (Srivastav et al., 2024) | 68.21 | 99.60 | 54.82 |
| DisPOSE (Ours) | $\mathbf{81.49}^{\pm 1e\text{-}4}$ | $\mathbf{99.90}^{\pm 0.0}$ | $\mathbf{35.40}^{\pm 8.7e\text{-}3}$ |

**Pose Regression Module (Stage II).** In Tab. 6, we evaluate the effectiveness of our proposed hypergraph decoder by comparing it against the established 3D CNN-based V2V-Net (Moon et al., 2018) (used in VoxelPose (Tu et al., 2020) and SelfPose3d (Srivastav et al., 2024)) and the transformer-based MVGFormer (Liao et al., 2024). While V2V-Net shows limited accuracy at strict precision thresholds ($AP_{25}$ of 48.69), MVGFormer demonstrates improved fine-grained localization ($AP_{25}$ of 62.48). However, MVGFormer falls behind V2V-Net at less strict thresholds ($AP_{50}$ of 95.12 vs. 97.08). Our module overcomes these trade-offs, outperforming both baselines across all metrics.

**Diffusion Modeling.** As shown in Tab. 7, our proposed diffusion module yields the best root localization performance. While direct regression and unconstrained diffusion achieve similar MDE scores (35.5 mm), our method demonstrates

*Table 6.* Ablation study on the **regression module (stage II)**. We compare our hypergraph decoder against V2V-Net and the transformer-based MVGFormer.

| Module | AP (mm) ($\uparrow$) | | | | Recall ($\uparrow$) | MPJPE |
|---|---|---|---|---|---|---|
| | 25 | 50 | 100 | 150 | @500 | ($\downarrow$) |
| V2V-Net (Moon et al., 2018) | 48.69 | 97.08 | 99.16 | 99.37 | 99.73 | 24.49 |
| MVGFormer (Liao et al., 2024) | 62.48 | 95.12 | 98.47 | 99.23 | 99.58 | 23.16 |
| Hypergraph Decoder (Ours) | **68.58** | **98.59** | **99.60** | **99.80** | **99.91** | **21.20** |

superior precision, particularly at fine-grained thresholds. We achieve an $AP_{25}$ of 5.36 and an $AP_{50}$ of 86.01, outperforming both alternative approaches.

*Table 7.* Ablation study on our diffusion module for **assignment approximation (stage I)** on CMU Panoptic. We compare our diffusion-based approach against a standard direct regression approach, and an unconstrained diffusion approach.

| Method | AP-Root (mm) ($\uparrow$) | | | | Recall ($\uparrow$) | MDE |
|---|---|---|---|---|---|---|
| | 25 | 50 | 100 | 150 | @500 | (mm) $\downarrow$ |
| Direct Regression | 4.11 | 85.08 | 97.73 | 98.10 | 99.80 | 35.57 |
| Unconstrained Diffusion | 3.43 | 82.59 | 98.66 | 99.50 | **99.90** | 35.44 |
| Projected Diffusion (Ours) | **5.36** | **86.01** | **99.40** | **99.74** | **99.90** | **35.40** |

**Impact of Training Data Size.** In Tab. 8, we show that our method is highly data-efficient. Training on merely 9.8% of the data yields an $AP_{50}$ of 84.05, competitive with the full dataset. While SelfPose3D (Srivastav et al., 2024) achieves higher $AP_{25}$ by leveraging a synthetic catalog of 3D rootposes for explicit 3D supervision, our approach relies on triangulation, which limits millimeter precision, yet remains significantly more robust at coarser thresholds.

*Table 8.* Ablation study on training data size of **root regression (stage I)**, evaluting the performance change when the number of training sequences decreases on CMU Panoptic. We compare with SelfPose3D, which is supervised by a synthetic catalogue of 3D poses.

| Training Data | AP-Root (mm) ($\uparrow$) | | | | Recall ($\uparrow$) | MDE |
|---|---|---|---|---|---|---|
| | 25 | 50 | 100 | 150 | @500 | (mm) $\downarrow$ |
| 1 Sequence (9.8%) | 4.68 | 84.05 | 99.23 | 99.59 | 99.77 | 36.20 |
| 2 Sequences (12.4%) | 4.51 | 84.53 | 99.16 | 99.60 | 99.76 | 36.56 |
| 3 Sequences (14.8%) | 4.66 | 85.17 | 99.30 | 99.62 | 99.79 | 35.93 |
| 9 Sequences (100%) | 5.36 | 86.01 | 99.40 | 99.74 | 99.90 | 35.40 |
| Synthetic Catalogue (SelfPose3D) | 7.38 | 38.03 | 93.46 | 98.65 | 99.60 | 54.82 |

See the appendix for additional quantitative (Sec. D.1), qualitative (Sec. D.2), and ablation (Sec. D.3) results.

## 5. Concluding Remarks

We introduced DisPOSE, a self-supervised framework that effectively *disposes* of the need for 3D ground-truth supervision by formulating multi-view person assignment as a generative diffusion process on polystochastic tensors. We approximate discrete assignments via projected diffusion and employ a hypergraph-based decoder to accurately regress

geometrically and anatomically consistent 3D human poses. DisPOSE outperforms existing self-supervised methods on several standard benchmarks. Moreover, we show that Dis-POSE generalizes effectively to unseen camera arrangements and maintains robust performance on the challenging scenes of the proposed MM-OR POSE dataset. By bridging discrete assignment and continuous modeling, DisPOSE reliably resolves geometric ambiguities, demonstrating the potential of self-supervised learning for safety-critical applications where 3D ground truth is impractical. Code will be released at https://github.com/wngTn/DisPOSE.

**Limitations.** While effective, the current method requires preprocessing for pseudo-label generation and does not utilize available temporal information. Furthermore, our proposed MM-OR Pose benchmark demonstrates that extreme occlusions and out-of-distribution poses remain challenging for self-supervised methods; an individual visible in only a single view affords no further observations to triangulate, and consequently cannot be reconstructed. Moreover, extending our assignment framework to leverage temporal information remains a promising avenue for future work.

## Impact Statement

This work advances self-supervised multi-view 3D human pose estimation, eliminating the dependency on expensive 3D ground-truth to enable scalable applications across numerous domains; we find healthcare and human-robot collaboration particularly promising.

While this work shares the broader ethical considerations of the computer vision domain regarding biometric profiling, the risk of surreptitious surveillance is arguably lower in the specific multi-view acquisition setting. Our framework relies on calibrated multi-view environments, essentially limiting its deployment to controlled settings. While increased data efficiency lowers the barrier to training, the need for synchronized and calibrated camera arrays makes mass deployment in uncontrolled public spaces unlikely.

## Acknowledgments

The authors are grateful for support from the UK AI Research Resource (AIRR) through grant 0251-4584-0945-1 and from the Excellence Strategy of local and state governments in Bavaria, Germany, as well as computational resources of the LRZ AI service infrastructure provided by the Leibniz Supercomputing Center (LRZ), the German Federal Ministry of Education and Research (BMBF), and the Bavarian State Ministry of Science and the Arts (StMWK). T. B. was supported by a UKRI Future Leaders Fellowship (MR/Y018818/1). L.B. was supported by the UK Royal Society through grant NIF/R1/254128.

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

This appendix provides further details regarding our methodology, implementation, and experimental evaluation of DisPOSE. Sec. A elaborates on the algorithmic specifics, including the projected reverse diffusion process, loss functions, network architecture, and data augmentation strategies. Sec. B details our experimental setup, elucidating the construction of the MM-OR Pose dataset, dataset splits, and evaluation protocols for generalization experiments. Sec. C offers an extended discussion of related work. Finally, Sec. D presents additional quantitative and qualitative results, including ablation studies on supervision sources and computational resource usage.

## A. Methodology Details

### A.1. Projected Reverse Diffusion

In alg. 1, we provide the pseudocode for our projected reverse diffusion sampling procedure introduced in Definition 3.2 of the main text.

---

**Algorithm 1** Projected Reverse Diffusion (Sampling)

---

1: **Input:** Hypergraph $\mathcal{G}$, Trained denoiser $f_\theta$.
2: **Output:** Feasible assignment $\mathcal{X}_0 \in \mathcal{S}^{(V)}$.
3: Sample noise $\mathbf{u}_T \sim \mathcal{N}(0, \mathbf{I})$.
4: **Init:** $\mathcal{X}_T \leftarrow \Pi_\mathcal{S}(\mathbf{u}_T)$; $\mathbf{u}_T \leftarrow \phi(\mathcal{X}_T)$.
5: **for** $t = T, \ldots, 1$ **do**
6: $\quad$ Predict clean scores: $\hat{\mathbf{u}}_0^{\text{raw}} \leftarrow f_\theta(\mathcal{X}_t, t)$.
7: $\quad$ **Self-Correction:** Project prediction to manifold:
8: $\quad\quad$ $\hat{\mathcal{X}}_0 \leftarrow \Pi_\mathcal{S}(\hat{\mathbf{u}}_0^{\text{raw}})$
9: $\quad\quad$ $\hat{\mathbf{u}}_0 \leftarrow \phi(\hat{\mathcal{X}}_0)$ $\qquad\qquad\qquad\qquad\qquad\qquad$ // Embed $\hat{\mathcal{X}}_0$ to guide noise estimation
10: $\quad$ **Estimate Noise:**
11: $\quad\quad$ $\hat{\boldsymbol{\epsilon}}_t \leftarrow (\mathbf{u}_t - \sqrt{\bar{\alpha}_t}\hat{\mathbf{u}}_0)/\sqrt{1 - \bar{\alpha}_t}$
12: $\quad$ **Update (DDIM):**
13: $\quad\quad$ Compute $\mathbf{u}_{t-1}$ using Eq. (10).
14: $\quad$ **Manifold Projection:**
15: $\quad\quad$ $\mathcal{X}_{t-1} \leftarrow \Pi_\mathcal{S}(\mathbf{u}_{t-1})$
16: $\quad\quad$ $\mathbf{u}_{t-1} \leftarrow \phi(\mathcal{X}_{t-1})$ $\qquad\qquad\qquad\qquad\qquad$ // Coordinate transformation of $\mathcal{X}$
17: **end for**
18: **return** $\mathcal{X}_0$.

---

### A.2. Regression Supervision (Stage II)

In this section, we provide a detailed overview of the supervision strategy and loss functions introduced in Sec. 3.3 (main text) for the pose regression. We use an off-the-shelf 2D pose detector (Xu et al., 2022) to provide pseudo 2D keypoint labels. We train our pose regression network (Stage II) using a deep supervision scheme, applying losses at the output of every decoder layer $t$ (Liao et al., 2024). We formulate the training objective as a balance between data fidelity (fitting the 2D detector outputs) and geometric regularization (enforcing 3D consistency and validity).

**2D Supervision for Data Fidelity**. The primary signal comes from $\hat{\mathcal{Q}} = \{\hat{q}_{v,k}\}$, which denotes the set of 2D pseudo-poses detected in each view $v$ for person $k$. For a predicted 3D pose $P^{(\tau)}$ at layer $\tau$, we project the joints onto the image planes via the known camera projection functions $\pi_v(\cdot)$.

To balance precision with robustness to occlusion, we employ two complementary losses:

1. **Coordinate Loss ($\mathcal{L}_{\text{coord}}$):** we minimize the weighted $L_1$ distance between the projected points and the 2D detections. As defined in Eq. (16) in the main paper, we weight this loss by the 2D detector's confidence scores $s_{v,k,j}$, ensuring the model focuses on clearly visible keypoints while ignoring low-confidence noise:

2. **Heatmap Loss ($\mathcal{L}_{\text{hm}}$):** to allow the model to recover occluded joints without penalty, we employ an asymmetric loss on the heatmaps. Let $\mathbf{H}_{\text{pseudo}}$ be the Gaussian heatmap (Iskakov et al., 2019) generated from the 2D pseudo labels and $\mathbf{H}_{\text{pred}}^{(\tau)}$

be the heatmap rendered from the projected predictions at step $t$. We define the asymmetric heatmap loss as:

$$\mathcal{L}_{\text{hm}}^{(\tau)} = \sum \text{ReLU}(\mathbf{H}_{\text{pseudo}} - \mathbf{H}_{\text{pred}}^{(\tau)})^2. \tag{17}$$

This effectively penalizes the model if it misses an existing joint, but incurs no penalty if it predicts a valid joint that the off-the-shelf detector missed due to obstructions.

**Geometric Regularization**. We constrain the solution space using complementary regularization terms to enforce internal geometric consistency. To this end, we leverage the known camera parameters to generate 3D weak labels $\mathcal{P}_{\text{pseudo}}$ by triangulating 2D pseudo-labels using the predicted correspondences from Stage I.

1. **3D Anchor Regularization ($\mathcal{L}_{\textbf{anchor}}$):** we supervise the predicted 3D poses against the triangulated weak-labels $\mathcal{P}_{\text{pseudo}}$. While $\mathcal{P}_{\text{pseudo}}$ is noisy, it represents the geometric consensus of the multi-view system. We minimize $\|\mathcal{P}^{(\tau)} - \mathcal{P}_{\text{pseudo}}\|_1$ as a regularizer that anchors the network to the global coordinate system, preventing it from drifting into geometrically invalid configurations.

2. **Cross-Affine Consistency ($\mathcal{L}_{\textbf{affine}}$):** Following (Srivastav et al., 2024), we enforce that the predicted 3D geometry is invariant to camera frame perturbations. We apply random affine transformations to the input views and penalize discrepancies between the canonical pose predictions via an $\ell_1$ loss.

3. **Triangulation Residual Loss ($\mathcal{L}_{\textbf{tr}}$):** To enforce strict geometric validity of the predicted 2D offset positions, we minimize the smallest singular value of the triangulation constraint matrix, as proposed by (Zhao et al., 2023):

$$\mathcal{L}_{\text{tr}}^{(\tau)} = \sigma_{\min}\left(\mathbf{M}(\mathbf{p}_{\text{pred}})\right)^2. \tag{18}$$

where $\mathbf{M}(\mathbf{p}_{\text{pred}})$ is the measurement matrix constructed from the predicted 2D positions $\mathbf{p}_{\text{pred}}$ and camera projection matrices.

### A.2.1. TOTAL LOSS

The final objective combines the correspondence loss with the regression terms, where the latter are summed over all $T$ decoder layers:

$$\mathcal{L}_{\text{total}} = \lambda_{\text{diff}}\mathcal{L}_{\text{diff}} + \sum_{\tau=1}^{T}\left(\underbrace{\lambda_{\text{coord}}\mathcal{L}_{\text{coord}}^{(\tau)} + \lambda_{\text{hm}}\mathcal{L}_{\text{hm}}^{(\tau)}}_{\text{Data Fidelity}} + \underbrace{\lambda_{\text{anchor}}\mathcal{L}_{\text{anchor}}^{(\tau)} + \lambda_{\text{aff}}\mathcal{L}_{\text{affine}}^{(\tau)} + \lambda_{\text{tr}}\mathcal{L}_{\text{tr}}^{(\tau)}}_{\text{Geometric Regularization}}\right). \tag{19}$$

### A.3. Implementation Details

### A.4. Network Architecture Details

**Diffusion Denoiser Specifications**. The denoiser $f_\theta$ operates on the multi-view correspondence hypergraph $\mathcal{G} = (\mathcal{D}, \mathcal{E})$. We represent the diffusion state by explicit hyperedge weights $w_{t,e}$ (entries of $\mathcal{X}_t$). We extract node features $\{\mathbf{x}_i\}_{i \in \mathcal{V}}$ from intermediate backbone features (appearance evidence) and project them to a common hidden dimension $h_d = 256$. Hyperedge features are initialized from $w_{t,e}$, while the geometric cues $z_e$ are embedded and used for conditioning via affine modulation (Perez et al., 2018). We encode the timestep $t$ with a sinusoidal embedding and inject it into both node and edge streams. The model consists of 4 attention-based message passing layers on the bipartite incidence graph (nodes $\leftrightarrow$ hyperedges), alternating node$\rightarrow$edge and edge$\rightarrow$node updates with residual connections (Chien et al., 2022). A final set-to-edge MLP aggregates incident node features and combines them with the hyperedge embedding to output the scalar prediction $\hat{u}_{0,e}$.

**Decoder Specifications**. The decoder consists of $T = 4$ layers operating on learnable queries initialized from instance and joint-type embeddings (Wang et al., 2021). Each layer extracts visual features via projective attention (Wang et al., 2021). The refinement within each layer follows a topological sequence. First, a multi-view hypergraph convolution (Bai et al., 2021) refines features across views. Next, we predict 2D corrections and confidence scores for each projected joint using MLPs applied to the updated node features and perform differentiable algebraic triangulation (Iskakov et al., 2019) to obtain updated 3D coordinates. Finally, a person-part hypergraph convolution aggregates information across skeletal joints; these features are injected back into the stream to guide the subsequent layer.

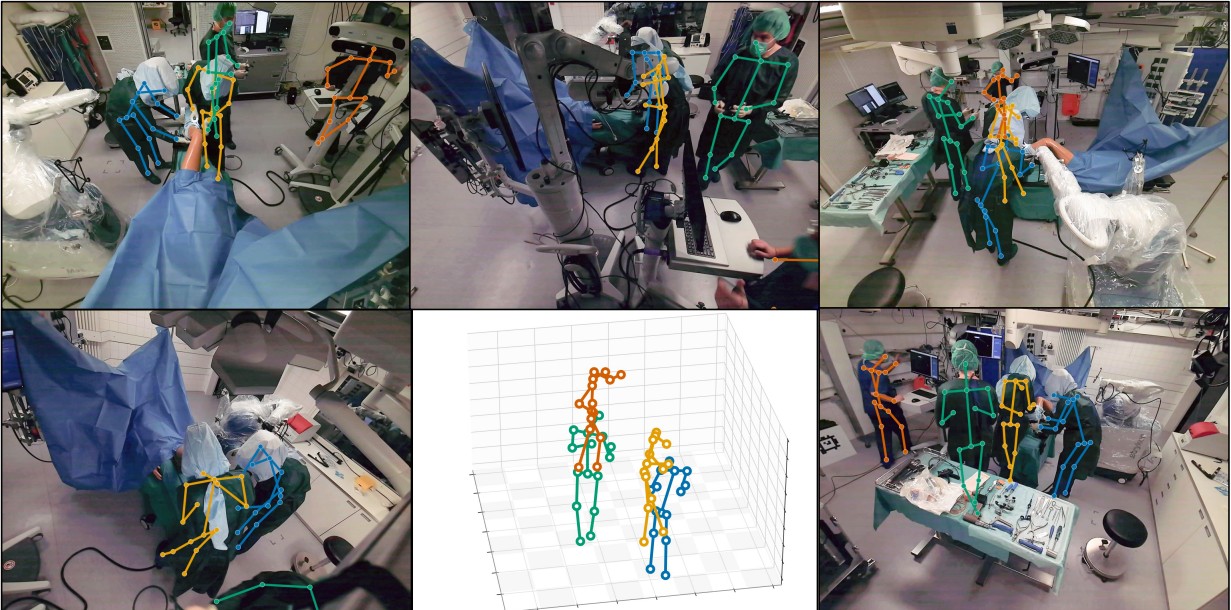

*Figure 6.* Sample frame from the proposed MM-OR Pose dataset with the manually annotated ground truth 3D human poses. **This example illustrates the heavy occlusions** and close interactions between individuals (the blue and yellow) in the operating room environment.

## A.5. Data Augmentation Details

Following the data augmentation protocol of (Srivastav et al., 2024), we apply geometric and photometric augmentations on the RGB input images during training. Geometric augmentations include random rotation sampled from $[-45°, 45°]$ and scaling from $[-0.35, 0.35]$. For photometric invariance, we apply spatial augmentations via RandAugment, which applies a random sequence of transformations including contrast, color, sharpness, and brightness jittering, as well as auto-contrast and equalization. Additionally, we apply RandCutout, which randomly masks out square masks of size $[20, 40]$ pixels across the image to simulate occlusions.

# B. Experimental Details

## B.1. Dataset Details

### B.1.1. MM-OR POSE

The original MM-OR (Özsoy et al., 2025) dataset contains 17 full-length (90 Minutes) videos of simulated robotic total and partial knee replacement surgeries. These surgeries were performed in a realistic surgical simulation environment, i.e., with realistic gowns, smocks, headwear, and medical equipment. The dataset contains various modalities, including RGB, depth, segmentation masks, audio and speech transcripts, robotic system logs, and infrared tracking. For our task, we use the five calibrated RGB videos from the multi-view RGB-D video stream from room cameras as input.

Compared to other datasets used for multi-view multi-human 3D pose estimation, MM-OR Pose contains a much higher degree of occlusions and more out-of-distribution poses, such as kneeling, crouching, bending, reaching over, and working in proximity to each other. Choudhury et al. (Choudhury et al., 2023), for instance, propose a subset of EgoHumans as an evaluation dataset for multi-view multi-human 3D pose estimation. However, EgoHumans depict individuals in an obstruction-free environment, with little to no occlusions or crowdedness.

In Figs. 6 to 8 we provide sample images from the MM-OR Pose dataset, illustrating the challenging nature of the operating room environment. In Fig. 6, we show an example frame with close interactions between two individuals, resulting in heavy occlusions. In Fig. 7, we show an example frame with an individual kneeling on the ground while working, illustrating the unusual poses contained in the dataset Finally, in Fig. 8 we show an example frame illustrating the annotation limitations that we encountered.

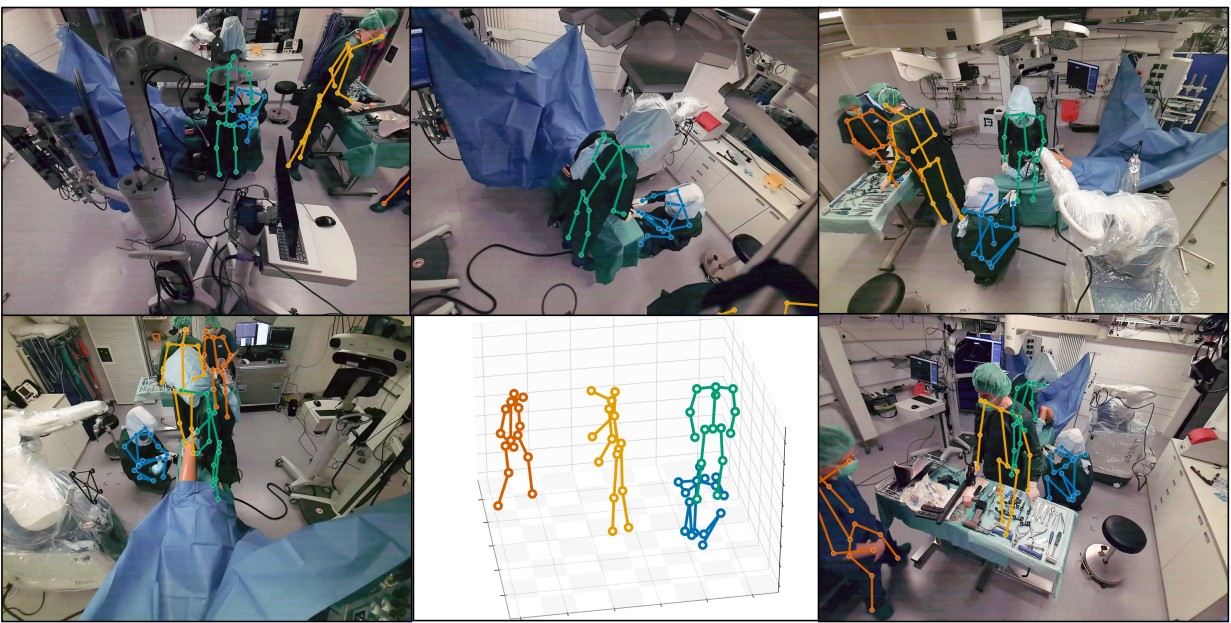

*Figure 7.* Sample frame from the proposed MM-OR Pose dataset with the manually annotated ground truth 3D human poses. **This example illustrates the unusual poses** contained in the dataset, showing the blue individual kneeling on the ground while working.

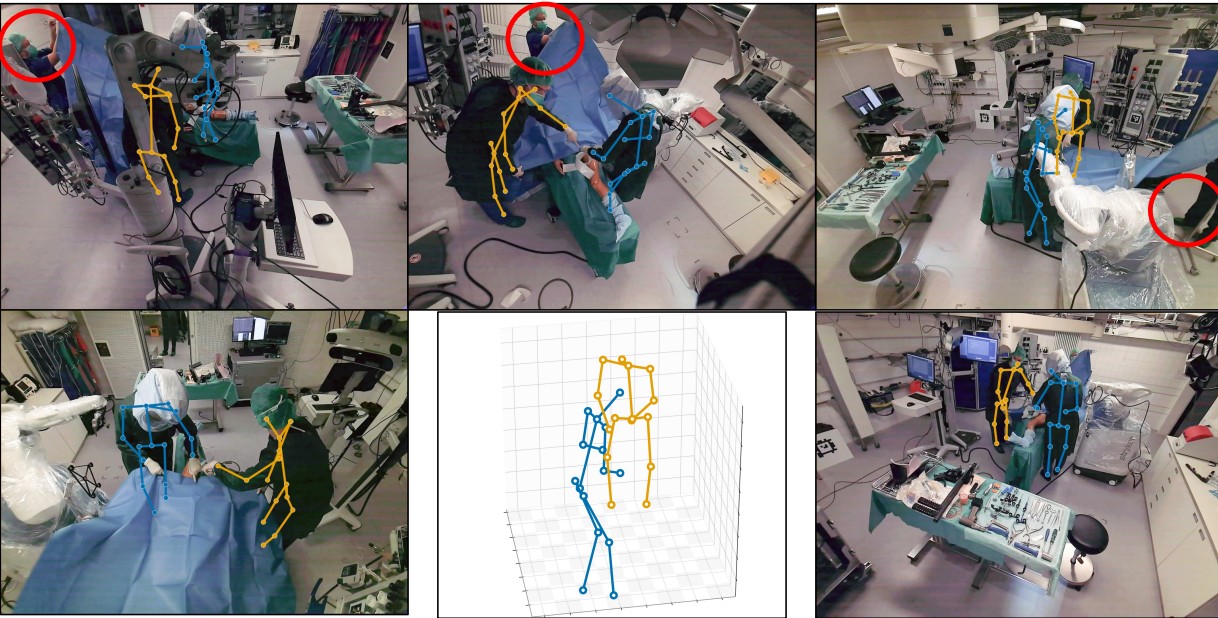

*Figure 8.* Sample frame from the proposed MM-OR Pose dataset with the manually annotated ground truth 3D human poses. **This example illustrates the annotation limitations** encountered during the annotation process. The red-circled individual is too heavily occluded to acquire reliable anchor poses for annotations. Additionally, that individual is positioned outside the depth sensor's field of view, making it profoundly difficult to accurately annotate the 3D pose.

**Annotation**. The MM-OR dataset does not contain any ground-truth 3D human pose annotations. As such, we manually annotated the test split (004_PKA, 011_TKA, 036_PKA, 038_TKA) of the dataset for evaluation purposes. To create anchor poses for annotation, we input the provided 2D human segmentation masks of MM-OR to SAM-3D body (Yang et al., 2025), to generate initial 3D human poses from each monocular view. We then aggregate the monocular 3D poses across all views into a common world, using correspondences from the segmentation masks.

We then manually refine the generated 3D poses using a custom annotation tool, which we implemented by adopting the 3D bounding box annotation tool (3D BAT) introduced by (Zimmer et al., 2019). We augment the original tool to support 3D human pose annotation and the real-time visualization of projected 2D keypoints in each RGB view.

As it is difficult to accurately annotate 3D human poses from five camera views, we also project the depth maps provided in MM-OR into a fused 3D point cloud of the scene to better estimate the 3D locations of keypoints. In Fig. 9, we provide an illustration of our annotation tool. On the left, the user can see the five RGB camera views, with the projected 2D keypoints overlaid in real-time. In the main view, the user can see the aggregated 3D point cloud of the scene, along with the initial 3D human poses. The user can then select and drag individual keypoints in 3D to refine the pose based on the RGB views and the 3D point cloud.

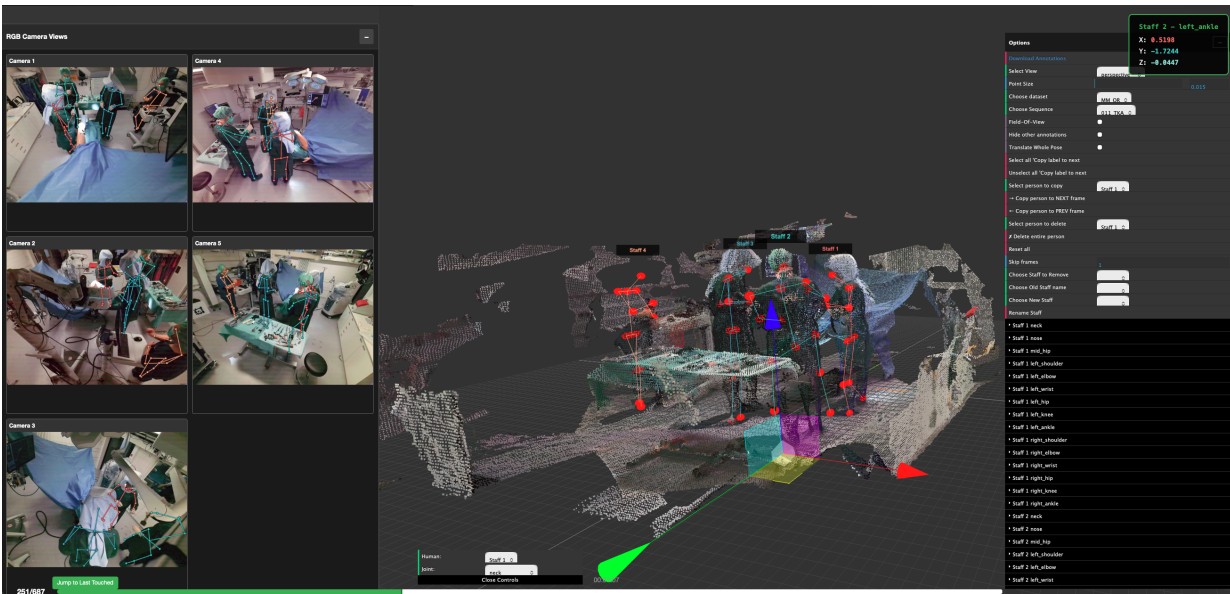

*Figure 9.* Screenshot of our custom 3D human pose annotation tool. On the left, the user can see the five RGB camera views, with the projected 3D keypoints overlaid. In the main view, the user sees the scene's 3D point cloud and the 3D human poses. The user selects and drags individual keypoints in 3D to refine the pose using both the RGB views and the 3D point cloud.

### B.1.2. CMU PANOPTIC

For the CMU Panoptic dataset (Joo et al., 2015), we follow the common practice of using the same training and evaluation splits as prior works (Tu et al., 2020; Ye et al., 2022; Wu et al., 2021; Srivastav et al., 2024; Liao et al., 2024). Specifically, we use the following sequences for training: 160422_ultimatum1, 160224_haggling1, 160226_haggling1, 161202_haggling1, 160906_ian1, 160906_ian2, 160906_ian3, 160906_band1, and 160906_band2. For evaluation, we use the following sequences: 160422_haggling1, 160906_pizza1, 160906_ian5, and 160906_band4. In line with prior work, we sample every 12th frame for evaluation, resulting in a total of 2,580 evaluation frames. For training and standard evaluation setup, we use the five cameras from the "CMU0" setup as defined in Tab. 9.

### B.1.3. SHELF DATASET

For the Shelf dataset (Belagiannis et al., 2014), we follow the training and evaluation splits as defined in prior works (Tu et al., 2020; Ye et al., 2022; Wu et al., 2021; Srivastav et al., 2024). We use the frames between index 300 to 600 for evaluation, and the remaining frames for training.

### B.1.4. CAMPUS DATASET

For the Campus dataset (Belagiannis et al., 2014), we follow the training and evaluation splits as defined in prior works (Tu et al., 2020; Ye et al., 2022; Wu et al., 2021; Srivastav et al., 2024). We use the frames between index 350 to 470, and 650 to 750 for evaluation, and the remaining frames for training.

### B.2. Pose Regression Evaluation on CMU Panoptic

We generally report the results of the methods as provided in their original publications. For Wu et al. (Wu et al., 2021) and MVGFormer (Liao et al., 2024), we additionally reproduced their results using their official codebases to obtain the Recall@500mm metric, for better comparability with the optimization-based and self-supervised methods.

### B.3. Root Regression Evaluation

In Tab. 5 of the main text, we evaluate the root regression module of VoxelPose (Tu et al., 2020), Faster VoxelPose (Ye et al., 2022), Wu et al. (Wu et al., 2021), and SelfPose3D (Srivastav et al., 2024). For VoxelPose, Faster VoxelPose, and SelfPose3D, the root positions are directly taken from their respective root regression module. For VoxelPose and SelfPose3D, this corresponds to their 3D CNN output; for Faster VoxelPose, to their 2D CNN + 1D height-estimation output. Wu et al. first perform pairwise correspondence matching to propose coarse root proposals. They subsequently refine these proposals by learning a neural distance function supervised by th e ground truth 3D root positions. In Tab. 5 we report the results with the refined root positions as "Wu et al. †", whereas "Wu et al. ‡" corresponds to the coarse root proposals before refinement. Because their pairwise matching produces many duplicate root proposals, we perform standard DBSCAN (Ester et al., 1996) clustering to obtain the final root positions for evaluation.

### B.4. Generalization on CMU Panoptic

We perform several generalization experiments on the CMU Panoptic dataset (Joo et al., 2015) to evaluate the effect of differing camera arrangements (see Tab. 4 of the main text) and camera numbers (see Tab. 10 in the appendix) on the performance of the baselines and our method. For training, we always use the CMU0 camera setup, as defined in Tab. 9, and perform evaluation *without* any fine-tuning. Here, we follow the proposed experimental setup of Liao et al. (MVGFormer) (Liao et al., 2024), which was later also adopted by Chharia et al. (MV-SSM) (Chharia et al., 2025). In Tab. 9, we summarize the different camera arrangements we use in our generalization experiments (see the main paper, Tab. 4 and Tab. 10, for details). Note that for the CMU4 setup, we use the first 4 cameras instead of the full 10 used in MVGFormer (Liao et al., 2024) to better align with camera setups that use fewer cameras.

*Table 9.* Definition of camera setups used in our ablation studies. We list the specific Camera IDs and the total number of views for each configuration on the CMU Panoptic dataset.

| Setup Name | Camera IDs | # Views |
|---|---|---|
| CMU0 | 3, 6, 12, 13, 23 | 5 |
| CMU0 w/ 2 extra | 3, 6, 12, 13, 23, 10, 16 | 7 |
| CMU0($K$) | First $K$ cameras from "CMU0 w/ 2 extra" | $K$ |
| CMU1 | 1, 2, 3, 4, 6, 7, 10 | 7 |
| CMU2 | 12, 16, 18, 19, 22, 23, 30 | 7 |
| CMU3 | 10, 12, 16, 18 | 4 |
| CMU4 | 6, 7, 10, 12 | 4 |

### B.5. Weak Supervision Labels

We obtain weak supervision labels for both the correspondence estimation and pose regression stages. For the 2D keypoints, we use an off-the-shelf 2D pose estimator (ViTPose (Xu et al., 2022)). To obtain cross-view correspondence labels to train our diffusion model, we use the higher-order correspondences predicted by COMPOSE (Wang et al., 2026). In Tab. 12, we also report the performance when using MvPose (Dong et al., 2019) to obtain weak labels. In this case, we use the output of their multi-view matching algorithm as pseudo-ground-truth correspondence labels. For both methods, we use the 2D keypoints predicted by the off-the-shelf 2D pose estimator as input.

### B.6. Training Differences with SelfPose3D (Srivastav et al., 2024)

SelfPose3D (Srivastav et al., 2024) proposes various techniques for their self-supervised training, including cross-affine-view consistency and L2 loss on projected heatmaps, which we also adopt in our method. However, SelfPose3D also uses a few additional training techniques that we do not adopt in our method:

- *Adaptive Supervision Attention:* SelfPose3D additionally trains a light-weight ResNet-18 to adaptively guide the supervision process based on the input images.

- *Curriculum Learning:* SelfPose3D increases the difficulty of training samples during their training, switching from soft labels with lower confidence to hard labels with higher confidence.

We show that even without these additional training techniques, our method still outperforms SelfPose3D.

## C. Related Works

Multi-view multi-human 3D pose estimation is now a well-established field with numerous solutions and sub-problem settings. We discuss closely related methods here and refer readers to a recent survey (Nogueira et al., 2025).

**Self-Supervised Methods**. Self-supervised methods have been extensively explored in monocular 3D human pose estimation and multi-view single-person 3D human pose estimation; however, they remain relatively underexplored in multi-view multi-human 3D pose estimation. In our task, (Rodriguez-Criado et al., 2024) achieves self-supervised estimation by curating a synthetic dataset with known correspondences and training a Graph Neural Network for (pairwise) cross-view matching using off-the-shelf 2D human poses. They subsequently use a re-projection loss to train an MLP that refines triangulated 3D coordinates by minimizing the distance between 2D detections and the projected 3D estimates. (Srivastav et al., 2024) propose an end-to-end self-supervised framework that jointly learns 2D detection and 3D pose estimation. However, they similarly create a synthetic catalog of 3D poses with known 3D ground truth to supervise the learning of their root regression module. (Liu & Zhang, 2025) propose a dense-sparse parallel network that additionally leverages temporal information. While achieving very strong performance, their method requires significant computational resources to incorporate temporal information (81 temporal frames) and falls short when fewer temporal frames are used. Overall, while considerable progress has been made in self-supervised multi-view multi-human 3D pose estimation, current methods still face challenges in computational efficiency, reliance on accurate 2D priors, and generalization to unseen camera setups.

**Fully-Supervised Methods**. Volumetric methods have established strong baselines by aggregating multi-view 2D features into voxelized 3D grids to estimate human poses (Iskakov et al., 2019; Tu et al., 2020; Chen & Tsai, 2025). However, these approaches require substantial GPU memory and computational resources due to their reliance on 3D CNNs. While efficient tri-planar approximations (Ye et al., 2022; Choudhury et al., 2023) mitigate these costs by reconstructing poses via orthogonal reprojection, they often degrade performance, particularly in crowded scenarios. Alternatively, plane sweep stereo techniques (Lin & Lee, 2021; de França Silva et al., 2022; Zhou et al., 2022) offer fast inference by projecting 2D joints into virtual depth planes; yet, they necessitate accurate 2D priors (e.g., ground truth bounding boxes), rendering them unsuitable for settings dependent solely on raw image inputs. Consequently, recent works have shifted toward explicit geometric representations to circumvent both volumetric overhead and strict prior dependencies. For instance, (Wang et al., 2021) proposes a transformer-decoder that iteratively refines 3D human poses from multi-view 2D features by projecting learnable 3D pose queries into each view. (Liao et al., 2024) builds upon this transformer-decoder architecture to improve generalization to unseen camera setups, by iteratively refining 2D offsets and triangulating 3D poses. (Chharia et al., 2025) advance this direction by integrating multi-view state-space models to explicitly capture joint spatial sequences. In our work, we also employ an iterative refinement strategy to refine 2D joint locations; however, we diverge by leveraging higher-order graph representations to explicitly model relational dependencies across views and individuals.

**Graph Modeling for Multi-View 3D Human Pose Estimation**. Graph Neural Networks have naturally become a cornerstone for modeling human skeletons, effectively capturing local kinematic constraints in monocular 2D (Yang et al., 2021) and 3D pose estimation (Zeng et al., 2021; Zou & Tang, 2021; Yu et al., 2023). However, extending these structures to multi-view multi-human settings presents a significant challenge, as the graph must expand to encompass not only intra-view body structures but also the explosive relation space of inter-view associations. To address this, (Wu et al., 2021) propose a Multi-view Matching Graph that formulates cross-view association via pairwise edge connectivity and learns to filter incorrect matches prior to triangulation. Yet, pairwise modeling often falls short in enforcing global consistency

across multi-view camera arrays, as errors in local dyadic decisions can propagate. Addressing this limitation, (Wang et al., 2026) recently introduced COMPOSE, shifting the paradigm toward higher-order representations, modeling 3D candidates as hyperedges within a hypergraph to capture holistic multi-view consensus via cover optimization. Building on this higher-order perspective, we adopt a similar graph formulation to explicitly model these complex dependencies; crucially, however, we move beyond fixed optimization solvers to learn these higher-order interactions directly, enabling our model to resolve global ambiguities end-to-end.

**Constrained Diffusion Models**. Diffusion models have gained significant traction across various areas of computer vision (Croitoru et al., 2023). While initial works primarily focused on image generation, recent studies have explored their application in solving generic geometric problems (Goren et al., 2025). However, the unconstrained nature of traditional diffusion models is often ill-suited to tasks that require strict adherence to fixed constraints. As such, a growing body of research focuses on constraining the diffusion process to satisfy specific constraints. (Fishman et al., 2023) propose log-barrier and reflected diffusion models to enforce hard constraints during the diffusion process. (Liu et al., 2023) proposes mirror diffusion models that, with the help of convex mirror functions, ensure that the diffusion dynamics remain in unconstrained space; however, the mirror function maps the samples back to constrained space. (Christopher et al., 2024) propose projection-based diffusion models that project the noisy samples back to the constrained space at each diffusion step. While theoretically sound, these constrained diffusion models require significantly more computational resources than traditional diffusion models. This limits their applicability for practical tasks. In this work, we propose a heuristic approach to constrain the diffusion process using projections, which remains computationally efficient while effectively satisfying the constraints of our task.

**Multi-View Synchronization**. *Synchronization* is defined as the recovery of absolute quantities from a collection of *ratios*, and is a fundamental component of classical multi-view pipelines (Govindu & Pooja, 2014; Govindu, 2004; Arrigoni et al., 2017; Wang & Singer, 2013; Thunberg et al., 2017; Tron & Vidal, 2014). Pioneering works use Lie group theory to average motions for Structure-from-Motion (SfM) problems (Govindu, 2004; Govindu & Pooja, 2014). Subsequent research extends these ideas to SE(3) via spectral decomposition and semidefinite programming, offering closed-form solutions and strong duality guarantees for global optimality (Arrigoni et al., 2016; Bernard et al., 2015; Chaudhury et al., 2015). To handle higher-order dependencies and heterogeneous collections, tensor-based approaches offer an alternative to flattened matrix formulations (Huang et al., 2019; Arrigoni & Fusiello, 2019). Other works focus on the uncertainty of these estimations; for instance, (Birdal & Simsekli, 2019) introduces probabilistic synchronization, while (Birdal et al., 2020) utilizes Sinkhorn divergences to synchronize point estimates as well as the distributions defined on them. Finally, quantum permutation synchronization advances discrete cases by solving non-convex optimization globally via quantum annealing on pairwise ratios (Birdal et al., 2021).

**Multi-Shape Matching and Generative Alignment**. While synchronization traditionally addresses rigid motions, recent approaches adapt these principles to non-rigid shape analysis and generative modeling. In the quantum domain, hybrid annealing methods decompose the NP-hard multi-shape matching problem into cycle-consistent triplets, scaling linearly with the number of shapes (Bhatia et al., 2023). To enforce cycle consistency in classical optimization, methods like MultiBodySync and SyNoRiM bootstrap dense correspondence without explicit labels (Huang et al., 2021; 2022; Bastian et al., 2024). Consistency can also be guaranteed by construction via shape-to-universe formulations (Gao et al., 2021), or refined explicitly using cycle consistency bases derived from directed graphs (Xia et al., 2025). A novel direction extends synchronization to *generative models*. SyncDiffusion synchronizes multiple joint diffusion processes via gradient descent on perceptual similarity losses (Lee et al., 2023), while SyncTweedies generalizes this to arbitrary generative distributions (Kim et al., 2024). Similarly, (Wu et al., 2024) adapt the diffusion processes to the doubly stochastic matrix space, treating correspondence estimation as a reverse denoising process that iteratively refines the matching matrix.

# D. Additional Results

## D.1. Additional Quantitative Results

In Tab. 10, we analyze the impact of varying the number of cameras during inference. In concordance with the generalization experiments in the main paper (Sec. 4.3), we train on the `CMU0` camera setup and evaluate on other camera setups without any fine-tuning. In contrast to the generalization experiments in the main paper, in this experiment, we do not change the camera arrangements but instead vary the number of cameras used during inference. DisPOSE shows consistent scalability, monotonically improving root and pose mAP as more cameras are added. However, in contrast, SelfPose3D (Srivastav et al., 2024) demonstrates instability with denser setups, with pose mAP dropping from 86.59% (4 views) to 78.77% (6 views).

Our proposed method effectively aggregates additional geometric evidence, achieving a peak pose mAP of 95.65% across 7 views.

*Table 10.* Ablation study for both **root regression (stage I) and pose regression (stage II)** on differing camera numbers. We compare SelfPose3d (Srivastav et al., 2024) and DisPOSE on variations of the CMU0 camera setup.

| Setup / Method | Root (Stage I) | | | Pose (Stage I & II) | | |
|---|---|---|---|---|---|---|
| | mAP (%) | Rec. (%) | MDE (mm)↓ | mAP (%) | Rec. (%) | MPJPE (mm)↓ |
| *CMU0 (3 views)* | | | | | | |
| SelfPose3d | 54.22 | 92.91 | 71.42 | 66.42 | 93.40 | 53.57 |
| DisPOSE (Ours) | **76.68** | **98.77** | **40.82** | **72.06** | **98.83** | **47.29** |
| *CMU0 (4 views)* | | | | | | |
| SelfPose3d | 62.29 | 99.44 | 63.98 | 86.59 | 99.44 | 28.99 |
| DisPOSE (Ours) | **79.26** | **99.83** | **39.36** | **92.13** | **99.85** | **23.83** |
| *CMU0 (6 views)* | | | | | | |
| SelfPose3d | 62.90 | 99.39 | 62.93 | 78.77 | 99.41 | 37.49 |
| DisPOSE (Ours) | **80.42** | **99.90** | **37.29** | **95.15** | **99.91** | **20.87** |
| *CMU0 (7 views)* | | | | | | |
| SelfPose3d | 63.02 | 99.24 | 62.58 | 78.73 | 99.20 | 37.11 |
| DisPOSE (Ours) | **80.41** | **99.90** | **37.34** | **95.65** | **99.91** | **20.49** |

### D.2. Qualitative Results

In Fig. 10 we present the qualitative example shown in the main paper (Fig. 4) with the original uncropped images for better context.

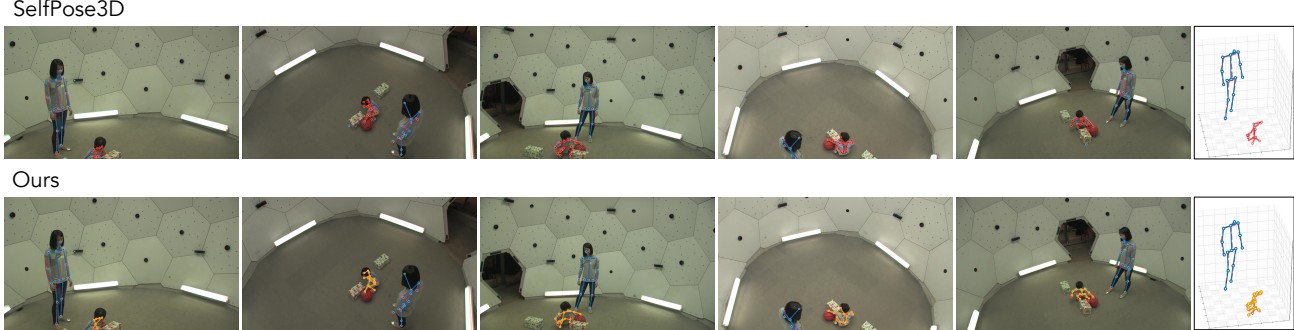

*Figure 10.* Qualitative *uncropped* example comparing SelfPose3D (Srivastav et al., 2024) and DisPOSE (Ours) on the CMU Panoptic dataset (Joo et al., 2015). Ground truth is shown in red, predictions in blue and orange. SelfPose3D fails to detect the toddler, likely due to domain shifts inherent in its synthetic training data. In contrast, DisPOSE successfully recovers the pose by learning structured correspondences directly from 2D priors.

In Fig. 11, we present the enlarged version of the qualitative example shown in the main paper (Fig. 5).

In Fig. 12, we provide an additional qualitative example from the MM-OR Pose dataset. This frame shows an individual kneeling next to the operating table, with an exceptionally low root joint position relative to the ground plane. This vertical translation causes SelfPose3D to fail completely, resulting in the subject being missed entirely. In contrast, DisPOSE successfully detects the subject and estimates the pose. However, the kneeling posture remains challenging; while detected, our method exhibits noticeable errors in the lower extremities when compared to the ground truth (see 3D view).

In Fig. 13, we show qualitative results on the Shelf dataset (Belagiannis et al., 2014). Our proposed method, DisPOSE, accurately estimates the 3D poses of all individuals in the scene, even when they are standing close to each other and

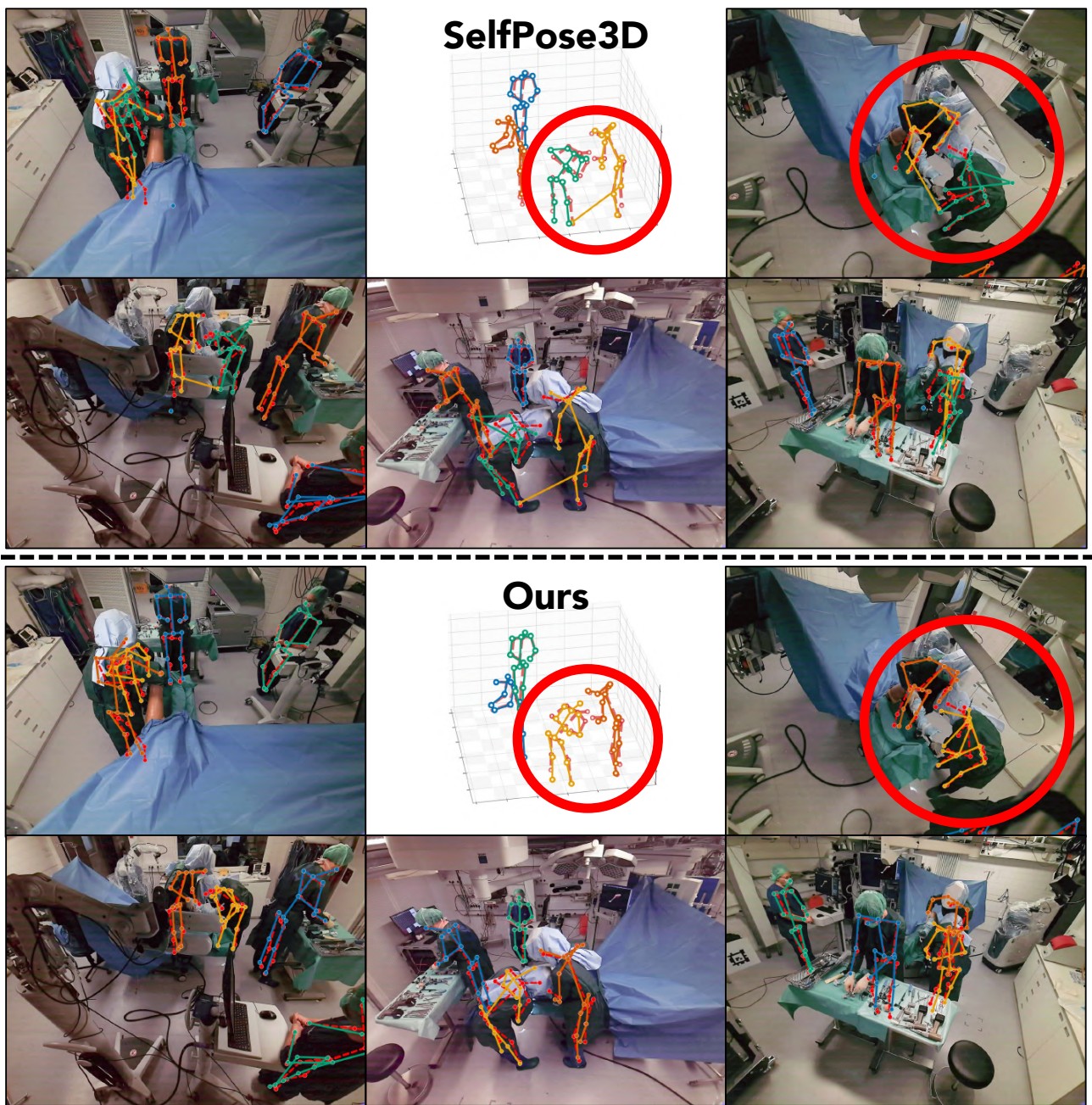

*Figure 11.* **Qualitative Example**. We compare SelfPose3D and DisPOSE (Ours) on the newly proposed MM-OR Pose dataset. Ground truth is shown in red, prediction in green, orange, yellow, and blue. Two surgeons are interacting closely together, while one is bent over the operating table, causing SelfPose3D to mispredict the left elbow position and the right ankle position as showin the red circles.

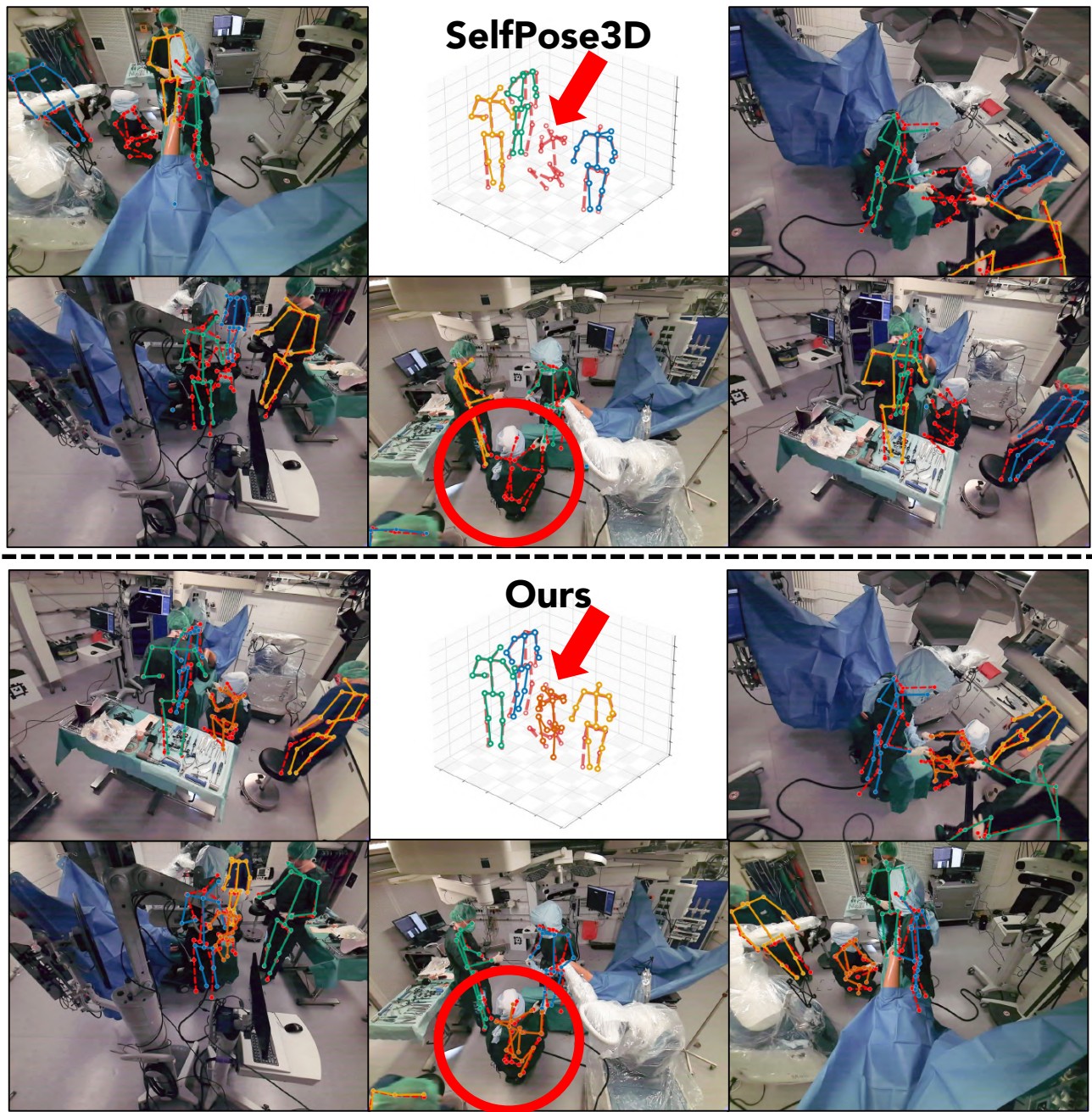

*Figure 12.* **Qualitative Example**. We compare SelfPose3D and DisPOSE (Ours) on the newly proposed MM-OR Pose dataset. Ground truth is shown in red, prediction in green, orange, yellow, and blue. One surgeon is kneeling next to the operating table. This unusual location causes SelfPose3D to miss the detection entirely, whereas DisPOSE can still estimate the pose.

partially occluded by the shelf.

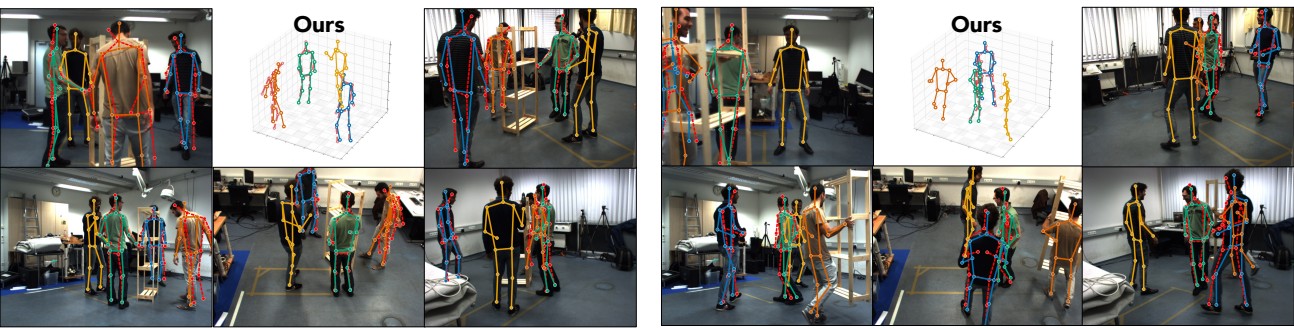

*Figure 13.* **Qualitative Example**. We show results on the Shelf dataset (Belagiannis et al., 2014). Ground truth is shown in red, prediction in green, orange, yellow, and blue. Our method accurately estimates the 3D pose of multiple people in close proximity, partially occluded by the shelf.

### D.3. Additional Ablation Studies

**GT vs. Pseudo 2D Keypoints.** In Tab. 11 we compare the impact of supervision quality. While ground truth supervision provides a significant boost in fine-grained precision (68.59% $P_{25}$ vs. 93.00% $AP_{25}$), our self-supervised approach remains highly competitive at coarser thresholds. This suggests that the performance gap is primarily driven by the precision and systematic bias of the 2D pseudo-label supervision signal.

*Table 11.* Ablation study on the source of supervision for pose estimation (stage I & stage II) on CMU Panoptic (Joo et al., 2015). We compare the performance when training with ground truth (GT) versus our self-supervised pseudo GT.

| Supervision | AP (mm) (↑) | | | | Recall (↑) | MDE |
|---|---|---|---|---|---|---|
| | 25 | 50 | 100 | 150 | @500 | (mm) ↓ |
| Ground Truth (GT) | 93.00 | 98.71 | 99.63 | 99.73 | 99.81 | 14.67 |
| Pseudo GT (Ours) | 68.59 | 98.59 | 99.60 | 99.80 | 99.91 | 21.20 |

**Optimization Engine For Pseudo Labels.** Tab. 12 demonstrates that our diffusion-based formulation for assignment approximation is robust to the source of pseudo-correspondence labels. Substituting our default engine (COMPOSE (Wang et al., 2026)) with MvPose (Dong et al., 2019) yields negligible performance differences (e.g., 35.44 mm vs. 35.40 mm MDE), implying that our framework generalizes well and does not rely on specific priors from a single optimization method. Moreover, when evaluated directly as a Stage-I solver at test time on the predicted heatmaps, DisPOSE (Stage I) surpasses

*Table 12.* Ablation study on the method that generates pseudo-correspondence labels used to train our diffusion model. We compare using MvPose (Dong et al., 2019) against COMPOSE (Wang et al., 2026) (our default).

| Engine | AP-Root (mm) (↑) | | | | Recall (↑) | MDE |
|---|---|---|---|---|---|---|
| | 25 | 50 | 100 | 150 | @500 | (mm) ↓ |
| MvPose (Dong et al., 2019) | 4.93 | 85.88 | 99.21 | 99.60 | 99.92 | 35.44 |
| COMPOSE (Wang et al., 2026) | 5.36 | 86.01 | 99.40 | 99.74 | 99.90 | 35.40 |

both pseudo-label engines on CMU Panoptic (see Tab. 13), showing that our learned model improves upon the teacher assignments rather than merely reproducing them.

**Stage-II Robustness to Root Errors.** In Tab. 14 we quantify the error propagation from Stage I to Stage II, by replacing the predicted 3D roots with ground-truth 3D roots and perturbed variants at inference time. MPJPE degrades by only 0.18 mm even with $\sigma = 50$ mm Gaussian noise on GT roots, and by 2.30 mm at $\sigma = 100$ mm. The marginal MPJPE increase with clean GT roots (+0.05 mm) is consistent with higher recall (99.98 vs. 99.91), indicating that additional, harder detections are recovered. This shows that our hypergraph decoder uses the triangulated root positions only for initialization, then iteratively refines joint positions by projecting them back into the 2D feature maps at each layer (Sec. 3.2).

*Table 13.* Direct Stage-I solver comparison on CMU Panoptic using the 2D root detections from the predicted heatmaps.

| Method | Root mAP (%) | Recall@500 (%) | MDE (mm)↓ |
|---|---|---|---|
| MvPose (Dong et al., 2019) | 76.67 | 99.90 | 38.63 |
| COMPOSE (Wang et al., 2026) | 80.35 | 99.84 | 36.29 |
| DisPOSE (Ours) | **81.49** | **99.90** | **35.40** |

*Table 14.* Robustness of Stage II to errors in the Stage-I root initialization on CMU Panoptic (Joo et al., 2015). We replace the predicted roots with ground-truth roots and noisy variants at inference time.

| Root Source | $AP_{25}$ (%) | $AP_{50}$ (%) | Recall@500 (%) | MPJPE (mm)↓ |
|---|---|---|---|---|
| Predicted (Stage I) | 68.59 | 98.59 | 99.91 | 21.20 |
| Ground Truth | 69.48 | 98.79 | 99.98 | 21.25 |
| Ground Truth + noise ($\sigma$=50 mm) | 67.82 | 97.39 | 99.85 | 21.38 |
| Ground Truth + noise ($\sigma$=100 mm) | 62.15 | 93.30 | 99.85 | 23.50 |

**Discrete Cross-View Association Errors.** We additionally quantify incorrect discrete cross-view associations, i.e., person swaps across views, by comparing each predicted hyperedge against ground-truth correspondences on the CMU Panoptic test set. As shown in Tab. 15, discrete miss-associations are rare, accounting for less than 1% of all predicted hyperedges. They mostly occur when people overlap heavily across several views, making the hyperedge cues ambiguous.

*Table 15.* Frequency of incorrect discrete cross-view associations on CMU Panoptic. We evaluate 8,831 predicted hyperedges in total, corresponding to approximately 3.4 hyperedges per frame on average.

| Metric | Value |
|---|---|
| Correctly associated *hyperedges* | 99.3% |
| *Frames* with $\geq$ 1 wrong association | 9.9% |

**Diffusion Time Steps and Sinkhorn Iterations.** In Tab. 16, we analyze the influence of the number of diffusion time steps ($T$) and Sinkhorn iterations ($L$) on the mean Average Precision (mAP) of the root regression task. Although the absolute differences are compressed due to averaging over six thresholds, distinct trends emerge. Increasing Sinkhorn iterations consistently yields performance gains, while the method proves efficient regarding diffusion time steps, achieving near-peak performance even with small $T$ To balance high performance with computational efficiency, we adopt T= 10 and $L = 4$ as our default configuration.

*Table 16.* Ablation on diffusion time steps ($T$) and Sinkhorn iterations ($L$). We report mAP (%) for each configuration.

| Sinkhorn Iterations ($L$) | Time Steps ($T$) | | |
|---|---|---|---|
| | 5 | 10 | 20 |
| 2 | 81.21 | 81.25 | 81.31 |
| 4 | 81.47 | 81.49 | 81.49 |
| 8 | 81.53 | 81.54 | 81.53 |

**Computational Resource Comparison.** In Tab. 17, we evaluate the computational efficiency of our method compared to the volumetric 3D CNN-based approach, SelfPose3D (Srivastav et al., 2024). We conduct experiments on a single NVIDIA A40 GPU (batch size 8) on the CMU Panoptic dataset (Joo et al., 2015). Our framework operates on the retained hyperedge support after geometric pruning, yielding modest increases in runtime and moderate increases in memory with increasing view count. Across all tested settings, DisPOSE is 2.8–4.6× faster than SelfPose3D; runtime increases from 452 ms at 5 views to 514 ms at 6 views, with peak memory reaching 4,521 MiB at 6 views.

*Table 17.* Computational resource comparison after replacing dense Sinkhorn projection with a sparse projection on the retained hyper-edge support.

| # Views | Inference Speed (ms) ↓ | | Peak GPU Memory ↓ | |
|---|---|---|---|---|
| | SelfPose3d | DisPOSE | SelfPose3d | DisPOSE |
| 3 | $1472^{\pm14}$ | $318^{\pm2.3}$ | 2156 MiB | 2370 MiB |
| 4 | $1450^{\pm33}$ | $376^{\pm2.4}$ | 2156 MiB | 3086 MiB |
| 5 | $1468^{\pm14}$ | $452^{\pm2.1}$ | 2156 MiB | 3804 MiB |
| 6 | $1445^{\pm23}$ | $514^{\pm3.1}$ | 2156 MiB | 4521 MiB |

*Table 18.* Component-wise profiling of DisPOSE at 5 and 6 views on CMU Panoptic.

| Module | 5 views | | 6 views | |
|---|---|---|---|---|
| | Latency (ms) | Share (%) | Latency (ms) | Share (%) |
| 2D Backbone | 182.8 | 40.3 | 197.9 | 38.5 |
| Hypergraph Decoder | 62.0 | 13.7 | 81.7 | 15.9 |
| Diffusion Denoise | 40.6 | 9.0 | 47.3 | 9.2 |
| Sinkhorn Projection | 12.6 | 2.8 | 15.4 | 3.0 |
| Other | 155.0 | 34.2 | 171.7 | 33.4 |
| Total | 453 | 100 | 514 | 100 |

**Component Profiling.** As shown in Tab. 18, the dominant cost stems from the shared 2D backbone and hypergraph decoder, while (sparse) Sinkhorn projection accounts for only 2.8–3.0% of runtime.

