# OpenReview forum: "DisPOSE: Projected Polystochastic Diffusion for Self-Supervised Multi-View 3D Human Pose Estimation"
_ICML.cc/2026/Conference — ICML 2026 regular_

### Official Review · Reviewer_knUt · 2026-03-05

**Soundness:** 3
**Presentation:** 3
**Significance:** 3
**Originality:** 3
**Overall Recommendation:** 5
**Confidence:** 4

**Summary:**

The paper presents DisPOSE, a method that casts multi-view person association as projected diffusion on polystochastic tensors with a hypergraph-based decoder for full-body 3D pose, achieving strong self-supervised results across Panoptic, Shelf/Campus, and a new OR benchmark.

**Compliance With Llm Reviewing Policy:**

Affirmed.

**Final Justification:**

While the paper’s technical foundation and originality were already quite strong, my initial hesitation about attributing the performance gains and the clarity of presentation was adequately resolved by the rebuttal’s ablations, sensitivity analyses, and concrete revision plans, leading to increase my score by one point.

**Key Questions For Authors:**

* How do the loss weights (e.g., $\lambda_{\text{diff}}$, $\lambda_{\text{coord}}$, $\lambda_{\text{hm}}$, $\lambda_{\text{geo}}$) and diffusion/Sinkhorn settings (steps $T$, iterations $L$) affect performance and stability across datasets?
* Can you provide component-wise causal ablations that isolate the benefits of (i) projected vs. unconstrained diffusion, and (ii) the hypergraph decoder vs. transformer/CNN baselines.
* Inference cost spikes at 6+ views. What is the complexity driver (tensor size, Sinkhorn iterations, incidence construction), and can you quantify the expected savings of a sparse Sinkhorn or block-diagonalization?
* How robust is Stage II to systematic detector biases (e.g., missed joints)?

**Limitations:**

The paper acknowledges several limitations.

**Strengths And Weaknesses:**

**Strengths**

* The formulation of assignment as diffusion projected via multi-marginal Sinkhorn is technically coherent.
*  Figures convey the two-stage pipeline clearly, and tables cover a wide range of baselines (fully supervised, optimization-based, self-supervised).


**Weaknesses**

* The first two paragraphs of the introduction are long and kind of philosophical, diluting focus; they could have been compressed to 1–2 sentences. The Preliminaries are overly long and dense, with multiple definitions that belong in the appendix. As written, Preliminaries outsize the Method section, which makes it not easy to follow the actual contribution. The symbol $P$ is used for both pose and assignment matrix, creating avoidable ambiguity. The Implementation Details (3.4) should have been included in the Experiments section.
* The improvement margins, while solid, are not dissected enough to attribute where they come from (projection vs. denoiser capacity vs. decoder design vs. supervision mix).

---

> ### Author Rebuttal · Authors · 2026-03-29
>
> *† marks new analyses conducted during the rebuttal period.*
>
> We appreciate the reviewer's careful reading and concrete suggestions on ablation clarity and presentation.
>
> **1. Component-wise causal ablations**
>
> The requested decomposition is already present across Tables 6, 7, 11, and 12 (the latter two in the Appendix). We summarize them here for convenience. All numbers are on CMU Panoptic, and in each sub-study, one factor is varied while the rest of the setup is kept fixed:
>
> | Component | Variant | Metric | Value |
> |---|---|---|---:|
> | Stage I formulation (Table 7) | Projected diffusion (Ours) | AP50 | **86.01** |
> |  | Unconstrained diffusion | AP50 | 82.59 |
> |  | Direct regression | AP50 | 85.08 |
> | Stage II decoder (Table 6) | Hypergraph (Ours) | MPJPE | **21.20 mm** |
> |  | V2V-Net (3D-CNN) | MPJPE | 24.49 mm |
> |  | MVGFormer (Transformer) | MPJPE | 23.16 mm |
> | 2D supervision (Table 11) | GT keypoints | MPJPE | 14.67 mm |
> |  | Pseudo-detected | MPJPE | 21.20 mm |
> | Pseudo-label source (Table 12) | COMPOSE | AP50 | **86.01** |
> |  | MvPose | AP50 | 85.88 |
>
> The projection step (Table 7) provides consistent improvement at the root level, especially at stricter thresholds (e.g., AP50: 86.01 vs. 82.59 over unconstrained diffusion). For the pose level, the larger gains come from the hypergraph decoder (Table 6: +1.96 to 3.29 mm over baselines) and from 2D supervision quality (Table 11: 6.53 mm MPJPE gap).
>
> **2. Hyperparameter sensitivity**
>
> The default hyperparameters were selected once on CMU Panoptic via a coarse grid search and then kept fixed for Shelf, Campus, and MM-OR Pose without dataset-specific retuning. Table 13 (Appendix) ablates $T = \{5, 10, 20\}$ and $L = \{2, 4, 8\}$; root mAP varies from 81.21 to 81.54. We additionally ran a one-at-a-time sensitivity analysis of the six loss weights (Sec A.2.1, Appendix) on CMU Panoptic at 0.5x and 2x the default, reporting pose mAP†:
>
> | Lambda varied | 0.5x† | 1x (default) | 2x† |
> |---|---:|---:|---:|
> | $\\lambda_\\text{diff}$ | 94.00 | 94.27 | 93.98 |
> | $\\lambda_\\text{coord}$ | 94.17 | 94.27 | 93.94 |
> | $\\lambda_\\text{hm}$ | 93.76 | 94.27 | 94.06 |
> | $\\lambda_\\text{anchor}$ | 94.26 | 94.27 | 93.07 |
> | $\\lambda_\\text{aff}$ | 93.50 | 94.27 | 94.24 |
> | $\\lambda_\\text{tr}$ | 93.80 | 94.27 | 93.76 |
>
> Performance drops by at most 1.2 points under 2x perturbation, indicating moderate sensitivity. The largest drop occurs when doubling $\\lambda_\\text{anchor}$ (-1.20 points). Using the same defaults across all four datasets also suggests that the settings are reasonably stable across datasets.
>
> **3. Inference cost at 6+ views — sparse Sinkhorn†**
>
> The original bottleneck was the dense Sinkhorn normalization, which materialized the full $\\mathcal{O}(N^V)$ tensor. During the rebuttal, we replaced it with a sparse solver that operates directly on the pruned hyperedge list. Since pruning retains only 0.12-0.25% of candidates at 5-7 views†, the cost scales as $\\mathcal{O}(L \\cdot V \\cdot \\#\\text{retained})$ rather than exponentially. Updated results†:
>
> | Views | DisPOSE Speed (ms) | DisPOSE Mem (MiB) | SelfPose3D Speed (ms) | SelfPose3D Mem (MiB) |
> |---|---:|---:|---:|---:|
> | 3 | 318 ± 2.3 | 2,370 | 1,472 | 2,156 |
> | 4 | 376 ± 2.4 | 3,086 | 1,450 | 2,156 |
> | 5 | 452 ± 2.1 | 3,804 | 1,468 | 2,156 |
> | 6 | 514 ± 3.1 | 4,521 | 1,445 | 2,156 |
>
> Sinkhorn now accounts for only 3.0% of runtime and 26 MiB at 6 views. DisPOSE is 2.8-4.6x faster than SelfPose3D across all view counts, and the previous 6-view spike is gone.
>
> **4. Robustness of Stage II to detector biases**
>
> Two design choices target this. First, the asymmetric heatmap loss (Appendix A.2) penalizes missed detections but not predictions where the detector failed, preventing the model from learning to suppress joints in occluded regions. Second, the confidence-weighted coordinate loss (Eq. 17) downweights low-confidence joints. Consistent with this, the pipeline remains strong with pseudo-2D supervision (Table 11; AP100: 99.60, only 0.03 below the GT keypoints).
>
> **5. Presentation and notation**
>
> We agree with both points. In the revised manuscript, we compressed the introduction, moved heavier preliminaries to the appendix, relocated the implementation details, and resolved the $P_k  / P_{ij}$ notation clash.
>
> ---
>
> We hope these consolidated ablations and clarifications make the sources of improvement, the sensitivity profile, and the remaining presentation changes substantially clearer and are helpful for the reviewer's final assessment.

---

> > ### Author Rebuttal · Reviewer_knUt · 2026-04-03
> >
> > I appreciate the authors’ detailed response. The rebuttal adequately addresses most of the concerns raised in my initial assessment. In addition, after reviewing the authors’ responses to the other reviewers’ comments, I have decided to increase the score by one point.

---

### Official Review · Reviewer_ScSv · 2026-03-10

**Soundness:** 3
**Presentation:** 2
**Significance:** 2
**Originality:** 2
**Overall Recommendation:** 4
**Confidence:** 2

**Summary:**

This paper introduces DisPOSE, a self-supervised multi-view 3D human pose estimation framework. It formulates the cross-view person assignment problem as a projected diffusion process over polystochastic tensors using Sinkhorn normalization, followed by a hypergraph decoder for 3D skeleton regression. A challenging surgical dataset, MM-OR POSE, is also proposed.

**Compliance With Llm Reviewing Policy:**

Affirmed.

**Final Justification:**

I am revising my recommendation from a "Weak Reject" to an "Accept" based on the authors' rebuttal.
In my initial review, I had raised two major concerns: the severe scalability bottleneck caused by the dense Sinkhorn implementation, and the uncertainty regarding whether the student model could surpass the teacher optimizer's performance.

The authors have convincingly addressed both issues:

Scalability: The introduction of the sparse Sinkhorn solver is a significant technical improvement. The new results show a drastic reduction in memory usage (from ~8GB to ~4.5GB at 6 views) and a manageable linear runtime increase. This effectively removes the previous barrier to practical deployment.

Performance: The new quantitative evidence (e.g., MPJPE of 21.20mm vs. COMPOSE's 23.62mm) demonstrates that the model can indeed learn to correct systematic errors in the teacher's pseudo-labels, validating the self-supervised learning process.

Given that the core limitations have been resolved and the methodological contribution is now more complete, I believe the paper meets the acceptance criteria.

**Key Questions For Authors:**

**Addressing the Bottleneck:** The memory explosion at 6+ views is a critical flaw. Do you have preliminary results or a concrete roadmap for a sparse tensor implementation of the Sinkhorn operator?

**Teacher-Student Bound:** Can you demonstrate specific instances or metrics where DisPOSE successfully corrects systematic assignment errors made by the teacher optimizer (e.g., COMPOSE) used for pseudo-labels?

**Temporal Feasibility:** How could temporal information be integrated into the current diffusion state without further exacerbating the existing memory limits?

I find the core idea very promising. **I am highly willing to raise my score to an Accept if the scalability bottleneck and the teacher-model dependency concerns are convincingly addressed during the rebuttal.**

**Limitations:**

Please refer to weaknesses and questions.

**Strengths And Weaknesses:**

### **Strengths:**

**Methodological Novelty: **Framing the discrete multi-view assignment as a projected diffusion process on polystochastic tensors is mathematically elegant.

**Generalization:** The method effectively removes the constraints of fixed volumetric setups, showing impressive robustness on unseen camera arrangements.

**Valuable Benchmark:** The MM-OR POSE dataset addresses a critical gap in heavily occluded, real-world multi-view scenarios.

### **Weaknesses:**


**Severe Scalability Bottleneck:** The dense Sinkhorn implementation causes an exponential memory spike. As shown in Table 14, Peak GPU Memory jumps from 2156 MiB (5 views) to 8254 MiB (6 views), severely limiting practical deployment.

**Dependency on Teacher Solvers:** The "self-supervised" claim is weakened by its reliance on classical optimizers (COMPOSE/MvPose) to generate correspondence pseudo-labels. It is unclear if the model can surpass the theoretical upper bound of these teacher algorithms.

**Lack of Temporal Modeling:** The framework ignores temporal dynamics, which are typically essential for resolving the extreme occlusions present in the authors' own MM-OR POSE dataset.

---

> ### Author Rebuttal · Authors · 2026-03-29
>
> *† marks new analyses conducted during the rebuttal period.*
>
> We appreciate the reviewer's comments, especially the positive feedback on the projected diffusion formulation, the cross-camera generalization results, and the MM-OR Pose benchmark. The review was also very clear about the two issues that matter most for reassessment.
>
> **1. Sparse Sinkhorn implementation removes the previous 6-view bottleneck†**
>
> The original bottleneck was the dense Sinkhorn normalization, which materialized a full $\\mathcal{O}(N^V)$ tensor even though pruning keeps only a tiny fraction of candidate hyperedges:
>
> | Views | Retained (%)† | Retained hyperedges† |
> |---|---:|---:|
> | 5 | 0.25 ± 0.01 | 87.8 ± 0.8 |
> | 6 | 0.16 ± 0.01 | 179.9 ± 1.6 |
> | 7 | 0.12 ± 0.01 | 357.2 ± 3.5 |
>
> We implemented a sparse Sinkhorn solver that works directly on the pruned hyperedge list.
> For each view, the required marginals are computed by grouping retained edges that share the same node in that view and applying grouped log-sum-exp with `scatter_reduce` and `scatter_add`. Each retained edge is therefore processed once per view in each Sinkhorn iteration, and as such the cost scales as $\\mathcal{O}(L \\cdot V \\cdot \\#\\text{retained})$ rather than exponentially with the number of views. The sparse solver computes the same projection on the pruned support.
>
> Updated inference cost with sparse Sinkhorn†:
>
> | Views | DisPOSE Speed (ms) | DisPOSE Mem (MiB) | SelfPose3D Speed (ms) | SelfPose3D Mem (MiB) |
> |---|---:|---:|---:|---:|
> | 3 | 318 ± 2.3 | 2,370 | 1,472 | 2,156 |
> | 4 | 376 ± 2.4 | 3,086 | 1,450 | 2,156 |
> | 5 | 452 ± 2.1 | 3,804 | 1,468 | 2,156 |
> | 6 | 514 ± 3.1 | 4,521 | 1,445 | 2,156 |
>
> DisPOSE is now 2.8-4.6x faster than SelfPose3D at all view counts. Runtime increases only 12% from 5 to 6 views (452 to 514 ms), versus 475 to 1,664 ms with the dense version. At 6 views, memory decreases from 8,254 to 4,521 MiB (-45%).
>
> Component profiling† confirms that the Sinkhorn bottleneck is resolved:
>
> | **Module** | **5v Latency (ms)** | **5v Share (%)** | **5v Mem (MiB)**  | **6v Latency (ms)** | **6v Share (%)** | **6v Mem (MiB)** |
> | :--- | ---: | ---: | ---: |  ---: | ---: | ---: |
> | **2D Backbone** | 182.8 | 40.3 | 3,364 | 197.9 | 38.5 | 4,037 |
> | **Hypergraph Decoder** | 62.0 | 13.7 | 1,183 | 81.7 | 15.9 | 1,418 |
> | **Diffusion Denoise** | 40.6 | 9.0 | 46 | 47.3 | 9.2 | 79 |
> | **Sinkhorn Projection** | 12.6 | 2.8 | 14 | 15.4 | 3.0 | 26 |
> | **Other** | 155.0 | 34.2 | 1,194 | 171.7 | 33.4 | 1,431 |
>
> Sinkhorn now accounts for only 3.0% of runtime and 26 MiB at 6 views. The dominant cost is the shared 2D backbone (38.5% / 4,037 MiB), common to methods using multi-scale features (e.g., MvGFormer).
>
> **2. DisPOSE can surpass the teacher at test time**
>
> End-to-end performance across benchmarks (Tables 1, 3):
>
> | Dataset | COMPOSE | MvPose | DisPOSE |
> |---|---:|---:|---:|
> | CMU Panoptic (MPJPE mm) | 23.62 | 26.46 | **21.20** |
> | Shelf (PCP %) | 96.2 | 96.9 | **97.1** |
> | Campus (PCP %) | **97.3** | 96.3 | 95.6 |
>
> DisPOSE surpasses both baselines on CMU Panoptic and Shelf. On Campus (220 frames, 3 cameras), COMPOSE retains an edge; the smaller setup both makes the assignment problem less ambiguous for the optimizer and limits the volume of pseudo-labels available for training.
>
> To isolate Stage I, we run COMPOSE and MvPose at test time on the same 2D heatmaps:
>
> | Method | Root mAP (%) | Recall (@500mm) | MDE (mm) |
> |---|---:|---:|---:|
> | MvPose† | 76.67 | 99.90 | 38.63 |
> | COMPOSE† | 80.35 | 99.84 | 36.29 |
> | **DisPOSE** | **81.49** | **99.90** | **35.40** |
>
> DisPOSE surpasses the teacher at the assignment level (+1.14† mAP over COMPOSE). Table 4 further shows generalization to four unseen camera layouts, indicating the model is not limited to reproducing the teacher.
>
> Following prior work such as SelfPose3D and DSP, we use "self-supervised" to mean that no ground-truth 3D annotations are used for training, and that supervision instead comes from 2D detections and optimization-derived pseudo-correspondences. We agree that this distinction should be stated more explicitly, and we have revised the manuscript wording accordingly.
>
> **3. Temporal modeling**
>
> The framework is intentionally per-frame to establish the geometric core; comparable methods (SelfPose3D, VoxelPose, and MvPose) are also per-frame. Temporal consistency is typically a different problem setting, as it also requires tracking individuals, which can be ambiguous on its own. A natural extension is to condition the denoiser on adjacent-frame assignments; since sparse Sinkhorn now consumes only 3.0% of runtime, this would not reintroduce the previous bottleneck. We view this as an important next step orthogonal to the current contribution.
>
> ---
>
> Given the two issues the reviewer highlighted, we hope the new sparse-Sinkhorn results and teacher-student analysis address them directly, and we would be grateful if the reviewer would reconsider their assessment in light of this evidence.

---

> > ### Author Rebuttal · Reviewer_ScSv · 2026-04-03
> >
> > have read the author rebuttal. My concerns have been adequately addressed. I am selecting (a) Fully resolved.
> >
> >
> > The authors have successfully addressed my main concerns and I am happy to increase my score.

---

### Official Review · Reviewer_4JDz · 2026-03-11

**Soundness:** 3
**Presentation:** 3
**Significance:** 2
**Originality:** 3
**Overall Recommendation:** 4
**Confidence:** 4

**Summary:**

The paper introduces a novel framework that treats the multi-view person-assignment problem as a generative diffusion process. It avoids the need for explicit 3D ground-truth supervision by diffusing over the space of polystochastic tensors and using Sinkhorn projections to enforce valid associations. The system is composed of two stages: a projected diffusion process for 3D root localization and a hypergraph-convolutional decoder for full 3D skeleton regression.

**Compliance With Llm Reviewing Policy:**

Affirmed.

**Key Questions For Authors:**

1. How does the choice of the number of Sinkhorn iterations impact the trade-off between inference speed and the geometric consistency of the recovered 3D roots?
2. If someone completely disappears in the view, how to handle it?
3. How sensitive is the hypergraph decoder to errors in the 2D root candidates?

**Limitations:**

Yes

**Strengths And Weaknesses:**

Strengths:
The paper proposes a self-supervised system that models discrete multi-view assignment through projected diffusion. A method is designed to solve higher-order correspondences using multi-marginal Sinkhorn normalization within a generative pipeline. In addition, a new dataset capturing complex, highly occluded surgical operating room environments is collected.

Weaknesses:
The proposed method still relies on the quality and accuracy of off-the-shelf 2D pose detectors and heatmaps. Identifying body joints remains difficult in cases of complex production environments. The multi-marginal Sinkhorn iterations are truncated (L iterations), which may introduce approximation errors in very dense or complex scenes.

---

> ### Author Rebuttal · Authors · 2026-03-29
>
> *† marks new analyses conducted during the rebuttal period.*
>
> We appreciate the reviewer's comments and the positive feedback on both the projected-diffusion formulation and the MM-OR Pose benchmark.
>
> **1. Reliance on 2D pose detector quality**
>
> We agree that 2D supervision quality is a remaining bottleneck. On CMU Panoptic, replacing pseudo-detected 2D keypoints with ground truth improves AP25 from 68.59 to 93.00 and MPJPE from 21.20 to 14.67 mm (Table 11, Appendix), bringing us within 0.69 mm of the best fully supervised result (Voxel.+3DSA, 13.98 mm). The gap is concentrated at strict thresholds, while coarser thresholds are nearly unchanged (AP100: 99.60 vs. 99.63), indicating that our multi-view formulation itself is already strong and that the remaining error largely stems from fine localization and detector bias. Importantly, strong 2D supervision is much easier to obtain at scale than 3D pose labels, and DisPOSE is designed to benefit directly from better 2D detectors without requiring any 3D ground truth.
>
>
> **2. Truncated Sinkhorn iterations: approximation error and speed/consistency tradeoff**
>
> Table 13 (Appendix) ablates $L = \{2, 4, 8\}$ across $T = \{5, 10, 20\}$. Increasing $L$ from 2 to 4 improves root mAP from 81.25 to 81.49 (+0.24), while increasing further to $L = 8$ yields only a marginal gain to 81.54 (+0.05). This indicates that $L = 4$ is already close to the converged regime on CMU Panoptic and provides the best accuracy-efficiency tradeoff.
>
> During the rebuttal period, we also implemented a sparse Sinkhorn solver that operates directly on the pruned hyperedge list instead of a dense $\\mathcal{O}(N^V)$ tensor. After pruning, only a tiny fraction of candidate hyperedges remains, namely 0.25%, 0.16%, and 0.12% at 5, 6, and 7 views, respectively†. Since the sparse operator computes the same marginals over the retained support, the Table 13 accuracy numbers remain unchanged, while the projection cost becomes small in practice:
>
> |  | L = 2 | L = 4 | L = 8 |
> |---|---:|---:|---:|
> | Root mAP (%) | 81.25 | 81.49 | 81.54 |
> | Sinkhorn Projection (ms)† | 8.74 | 12.6 | 21.80 |
>
> At the default $L = 4$, Sinkhorn projection takes 12.6 ms† at 5 views, a small fraction of the 452 ms total inference time. Overall, $L = 4$ captures nearly all of the geometric-consistency benefit while keeping Sinkhorn to a small fraction of the full pipeline.
>
> **3. Handling complete disappearances from a view**
>
> Each view includes a dustbin node that absorbs unmatched identities (Sec. 2, footnote on the assignment tensor). Hyperedges contain at most one detection per view (Definition 2.2), so a person can be absent from any subset of views. Partial visibility, including complete disappearance from one or more cameras, is therefore supported by design.
>
> **4. Sensitivity of the hypergraph decoder to 2D root errors**
>
> Table 6 evaluates three decoders (V2V-Net, MVGFormer, Ours) with identical Stage I root inputs, isolating the decoder's contribution. We additionally replace predicted 3D roots with ground-truth 3D roots†:
>
> | Root source | AP25 | AP50 | Recall@500 | MPJPE (mm) |
> |---|---:|---:|---:|---:|
> | Predicted (Stage I) | 68.59 | 98.59 | 99.91 | 21.20 |
> | Ground truth † | 69.48 | 98.79 | 99.98 | 21.25 |
> | Ground truth + noise ($\sigma$=25 mm)† | 68.23 | 97.78 | 99.85 | 21.27 |
> | Ground truth + noise ($\sigma$=50 mm)† | 67.82 | 97.39 | 99.85 | 21.38 |
> | Ground truth + noise ($\sigma$=100 mm)† | 62.15 | 93.30 | 99.85 | 23.50 |
>
> MPJPE degrades by only 0.18 mm even with $\sigma$=50 mm Gaussian noise on GT roots, and by 2.30 mm at $\sigma$=100 mm†. The marginal MPJPE increase with clean GT roots (+0.05 mm) is consistent with higher recall (99.98 vs. 99.91), indicating that additional, harder detections are recovered. This shows that our hypergraph decoder uses the triangulated root positions only as an initialization and then iteratively refines joint positions by projecting them back into the 2D feature maps at each layer (Sec. 3.2).
>
> ---
>
> We hope these clarifications resolve your questions on detector dependence, Sinkhorn truncation, missing views, and Stage II robustness and are helpful for your final assessment.

---

> > ### Author Rebuttal · Reviewer_4JDz · 2026-04-03
> >
> > I thank the authors for the rebuttal. This time I will maintain the score.

---

> > > ### Author Response · Authors · 2026-04-04
> > >
> > > We are pleased to see that all the reviewers' concerns have been fully resolved. As such, we kindly ask the reviewer to consider increasing their score accordingly. We remain fully available throughout the discussion period to answer any additional questions.
> > >
> > > To summarize: this paper addresses a fundamental gap for 3D pose estimation, an invaluable task for action recognition and other downstream tasks in safety-critical domains such as surgical operating rooms. Our approach addresses this in an innovative way by decoupling the assignment and triangulation problems and demonstrates impressive results, including robustness to new camera views.

---

### Official Review · Reviewer_vMJT · 2026-03-12

**Soundness:** 3
**Presentation:** 3
**Significance:** 3
**Originality:** 3
**Overall Recommendation:** 4
**Confidence:** 4

**Summary:**

This paper studies self-supervised multi-view 3D human pose estimation from calibrated RGB cameras. The problem is decomposed into two parts: cross-view person association and 3D skeleton regression. The authors model the discrete association step as a diffusion process over polystochastic tensors with differentiable Sinkhorn projections to enforce valid assignments guided by 2D poses. After matching individuals across views, a hypergraph-based decoder predicts the final 3D poses. Because the method mainly relies on geometric consistency rather than a fixed camera setup, it generalizes well to different camera configurations. The paper also introduces a new multi-view surgical benchmark with strong occlusions.

**Compliance With Llm Reviewing Policy:**

Affirmed.

**Ethical Review Concerns:**

N/A.

**Final Justification:**

I find the paper technically solid, with a well-motivated formulation and convincing generalization across camera setups. The rebuttal is helpful and addresses my main concerns, especially on scalability and robustness, which improves my confidence. Some issues remain partially resolved, including reliance on pseudo-labels, limited analysis of failure cases, and the relatively small benchmark. Overall, the rebuttal strengthens the paper but does not change my overall view, so I keep my original recommendation.

**Key Questions For Authors:**

Please see the **Weaknesses**.

**Limitations:**

Please see the **Weaknesses**.

**Strengths And Weaknesses:**

**Strengths**

**1. Principled formulation.**
The paper proposes a non-trivial formulation for cross-view person association by relaxing permutation tensors to the polystochastic tensor space and performing diffusion with Sinkhorn projections. This provides a principled way to enforce assignment constraints. Ablations (Table 7) suggest the projection step improves high-precision metrics (e.g., AP₂₅) compared with direct regression or unconstrained diffusion.

**2. Strong cross-camera generalization results.**
The zero-shot camera configuration experiments (Table 4) are particularly convincing. Prior methods such as SelfPose3d degrade significantly under new layouts (CMU2/CMU3), while DisPOSE remains relatively stable, indicating better robustness to varying camera setups.

**Weaknesses**

**1. Dependence on COMPOSE.**
The The pipeline appears to rely on COMPOSE. It is unclear whether comparable performance can be achieved with other publicly available methods (e.g., MvPose). Table 12 suggests a gap, but the paper does not discuss how sensitive the framework is to this component with more choices.

**2. Scalability with the number of views.**
The paper notes that complexity grows exponentially with the number of cameras, but practical implications are not analyzed. Inference time, memory usage for V≈5, and feasibility for larger setups (V≥6) are not reported.

**3. Error propagation in the two-stage design.**
Stage I predicts root positions, which initialize Stage II pose refinement. However, the paper does not analyze how Stage I errors (e.g., large outliers) affect the final MPJPE, leaving the robustness of the pipeline unclear.

**4. Limited scale of the MM-OR Pose benchmark.**
The dataset contains only 750 annotated frames from four surgical sequences, with acknowledged annotation difficulty under heavy occlusion. This raises questions about statistical reliability of the reported results.

**Minor issues**

- The gap to fully supervised methods is not discussed.

---

> ### Author Rebuttal · Authors · 2026-03-29
>
> *† marks new analyses conducted during the rebuttal period.*
>
> We appreciate the reviewer's thoughtful assessment, especially the positive feedback on the principled formulation and the cross-camera generalization results.
>
> **1. Dependence on COMPOSE for pseudo-label generation**
>
> We focus on COMPOSE and MvPose, the leading unsupervised association methods. Substituting COMPOSE with MvPose as pseudo-label engine yields negligible differences (Table 12; AP50 86.01 vs. 85.88). Despite MvPose producing noisier pseudo-labels (MDE 38.63 vs. 36.29), DisPOSE converges to nearly identical performance, suggesting the framework corrects rather than memorizes noisy labels. DisPOSE outperforms both baselines end-to-end (Tables 1, 3). Isolating Stage I:
>
> | Method | Root mAP (%) | Recall (@500mm) | MDE (mm) |
> |---|---:|---:|---:|
> | MvPose† | 76.67 | 99.90 | 38.63 |
> | COMPOSE† | 80.35 | 99.84 | 36.29 |
> | **DisPOSE** | **81.49** | **99.90** | **35.40** |
>
> DisPOSE surpasses both baselines at the assignment level (+1.14† mAP over COMPOSE), confirming that the improvement is not solely due to Stage II.
>
> **2. Scalability with the number of views**
>
> During the rebuttal, we replaced the dense Sinkhorn normalization with a sparse solver that operates on the pruned hyperedge list, scaling as $\\mathcal{O}(L \\cdot V \\cdot \\#\\text{retained})$ rather than $\\mathcal{O}(N^V)$ (see our response to Reviewer ScSv for details). Updated inference cost†:
>
> | Views | DisPOSE Speed (ms) | DisPOSE Mem (MiB) | SelfPose3D Speed (ms) | SelfPose3D Mem (MiB) |
> |---|---:|---:|---:|---:|
> | 3 | 318 ± 2.3 | 2,370 | 1,472 | 2,156 |
> | 4 | 376 ± 2.4 | 3,086 | 1,450 | 2,156 |
> | 5 | 452 ± 2.1 | 3,804 | 1,468 | 2,156 |
> | 6 | 514 ± 3.1 | 4,521 | 1,445 | 2,156 |
>
> DisPOSE is now 2.8-4.6x faster than SelfPose3D at all view counts. Runtime increases only by 12% from 5 to 6 views (452 to 514 ms), versus 475 to 1,664 ms with the dense version. Sinkhorn now accounts for only 3.0% of runtime at 6 views. The appendix camera-number ablation shows DisPOSE's pose mAP improves monotonically from 72.06% (3v) to 95.65% (7v), whereas SelfPose3D peaks at 86.59% (4v) and drops to 78.77% (6v), as their fixed voxel grid approach is anchored to the training camera configuration (Sec. 1).
>
> **3. Error propagation from Stage I to Stage II**
>
> Table 6 isolates Stage II under identical Stage I inputs; our decoder achieves the best MPJPE (21.20 vs. 24.49 / 23.16). To quantify root-error propagation, we feed GT roots into Stage II†:
>
> | Root source | AP25 | AP50 | Recall@500 | MPJPE (mm) |
> |---|---:|---:|---:|---:|
> | Predicted (Stage I) | 68.59 | 98.59 | 99.91 | 21.20 |
> | Ground truth† | 69.48 | 98.79 | 99.98 | 21.25 |
> | Ground truth + noise ($\sigma$=50 mm)† | 67.82 | 97.39 | 99.85 | 21.38 |
> | Ground truth + noise ($\sigma$=100 mm)† | 62.15 | 93.30 | 99.85 | 23.50 |
>
> MPJPE is essentially unchanged with GT roots (+0.05† mm; marginal increase reflects higher recall recovering harder detections). Even with $\sigma$=50 mm noise, MPJPE degrades by only 0.18 mm†. This suggests Stage I root noise is not the dominant limiter, and Stage II remains robust to fairly large root errors.
>
> **4. Limited scale of MM-OR Pose**
>
> The primary goal of MM-OR Pose is to evaluate pose estimation methods in challenging long-tail scenarios, such as surgical environments, for which annotated benchmarks are currently lacking. We designed this as a test-time evaluation benchmark, not as a large-scale training resource. For context, Shelf contains 300 frames and Campus 220; MM-OR Pose (750 frames) is larger than both combined. Annotations were verified via depth-fused point clouds (Appendix B.1.1). Per-sequence standard deviation†:
>
> | Metric | Overall | Per-seq. std† |
> |---|---:|---:|
> | AP50 (%) | 47.06 | 8.03 |
> | AP100 (%) | 83.59 | 0.32 |
> | MPJPE (mm) | 56.91 | 4.00 |
>
> AP100 and MPJPE are stable across sequences; AP50 varies more, as expected under severe occlusion, where tight thresholds amplify annotation noise. A larger surgical benchmark would warrant a dedicated dataset paper.
>
> **5. Gap to fully supervised methods**
>
> On CMU Panoptic, the MPJPE gap to the best fully supervised method is 7.22 mm (21.20 vs. 13.98; Table 1). The gap is concentrated at strict thresholds (AP25: 68.59 vs. 94.20); at AP100, we are already close (99.60 vs. 97.45-99.81 for supervised methods). Replacing pseudo 2D keypoints with GT keypoints narrows the MPJPE to 14.67 mm (Table 11), indicating that 2D supervision quality is the main bottleneck. On Shelf, our PCP (97.1%) is competitive with fully supervised methods (97.0–97.7%; Table 3). On Campus (95.6%), a small gap to the supervised range (96.2–97.3%) remains, but it is substantially reduced compared to prior self-supervised methods (DSP 92.8%, SelfPose3D 87.9%).
>
> ---
>
> We hope these analyses resolve the concerns regarding pseudo-label dependence, scalability, error propagation, benchmark scope, and the supervised gap and are helpful for your final assessment.

---

> > ### Author Rebuttal · Reviewer_vMJT · 2026-04-04
> >
> > Thx for the rebuttal addressing my main concerns with additional experiments, yet:
> >
> > **1. COMPOSE dependence:** MvPose results suggest robustness, but does not fully remove reliance on teacher distributions.
> >
> > ~**2. Scalability:** Sparse Sinkhorn seems to effectively resolve the practical bottleneck.~
> >
> > **3. Error propagation:** Robust to continuous noise; how about failures like a discrete one, incorrect cross-view associations? is that common & possible?
> >
> > **4. Dataset:** Reasonably justified but still limited in scale.
> >
> > ~**5. Supervised gap:** Attribution to 2D keypoints is plausible.~
> >
> > Overall, concerns are largely addressed in practice, while remaining issues may require more extensive analysis beyond rebuttal scope. Thus, I'll retain my initial scores.

---

> > > ### Author Response · Authors · 2026-04-06
> > >
> > > *† marks new analyses conducted during the rebuttal period.*
> > >
> > > We appreciate the follow-up and are pleased that our earlier responses addressed the concerns about scalability and the supervised gap.
> > >
> > > **1. COMPOSE dependence**
> > >
> > > DisPOSE surpasses both teacher baselines end-to-end (+1.14 mAP over COMPOSE; Tables 1, 3), which indicates that it improves on the teacher outputs rather than simply reproducing them. The teacher solvers use only 2D poses, without 3D annotations, and 2D quality is the main bottleneck (Table 11: GT keypoints reduce MPJPE from 21.20 to 14.67 mm).
> > >
> > > While foundation models for 2D pose continue to improve, 3D data and annotations are harder to obtain and thus much more scarce. Nevertheless, even in the long tail of very challenging 3D pose estimation environments, such as surgical operating rooms, this setting still stands to benefit from 2D priors. There are no viable alternatives for this problem setting — **it is ill-posed without additional structural information**. Selfpose3D additionally relies on 3D templates when training, but we show that this introduces bias (see Fig. 10, Sec. D.2).
> > >
> > > **3. Discrete cross-view association errors**
> > >
> > > We further evaluate incorrect discrete associations, i.e., person swaps across views, in terms of both their frequency and their downstream impact.
> > >
> > > **(a) Frequency.** We directly measure association accuracy by comparing each predicted hyperedge against GT correspondences on the CMU Panoptic test set (8,831 hyperedges, ~3.4 per frame on average)†:
> > >
> > > | Metric                                | Value |
> > > |:--------------------------------------|------:|
> > > | Correctly associated *hyperedges*     |99.3 % |
> > > | *Frames* with ≥1 wrong association    | 9.9 % |
> > >
> > > Discrete miss-associations are rare, accounting for <1% of all associations. They mostly occur when people overlap heavily across several views, making the hyperedge cues ambiguous.
> > >
> > > **(b) Downstream impact.** To test sensitivity beyond the observed error rate, we inject multiplicative noise into the hyperedge cues $z_e$ before denoising†. Note that this introduces a discrete change, swapping the hyperedge selected by the projected DDPM:
> > >
> > > | Noise $\sigma$ | Correct assoc. (%) | AP$_{25}$ | AP$_{50}$ | Recall@500 | MPJPE (mm) |
> > > |:--------------:|--------------------:|----------:|----------:|-----------:|-----------:|
> > > | 0.00           |                99.3 |     68.59 |     98.59 |      99.91 |      21.20 |
> > > | 0.10           |                94.8 |     68.46 |     98.47 |      99.85 |      21.22 |
> > > | 0.25           |                93.2 |     67.74 |     98.40 |      99.85 |      21.26 |
> > > | 0.50           |                91.4 |     67.38 |     98.10 |      99.74 |      21.33 |
> > >
> > > Even when correct associations drop from 99.3% to 91.4%, MPJPE increases by only **0.13 mm**. This matches our previous finding that Stage II is robust to continuous root perturbations (+0.18 mm at $\sigma$=50 mm Gaussian noise on GT roots). Wrong associations usually involve nearby people; as a result, the resulting root displacement remains limited, and Stage II can correct the residual error over successive refinement layers.
> > >
> > > **4. Dataset scale**
> > >
> > > Our goal in annotating sequences from MM-OR was to benchmark the particularly challenging long-tail scenario (in natural images) comprising baggy OR gowns, severe occlusions, and unusual poses. We thus annotated 750 evaluation frames across 3 sequences with different actors, which is larger than Shelf (300) and Campus (220) combined. Moreover, the sequence-level variation is low (MPJPE std: 4.00 mm), suggesting sufficient sample size. This is a valuable testing scenario for *self-supervised* methods, which has not been adequately explored.
> > >
> > > Extending MM-OR to a full-scale 3D pose benchmark with dedicated training and validation splits would be a valuable future direction, but this is beyond the scope of the present paper, as our contribution is primarily methodological. Annotating such OR sequences requires substantial manual effort, as keypoints must be refined in 3D to achieve adequate accuracy. To support such future extensions, we will also publicly release our custom annotation tool, which enables annotation refinement based on fused point clouds.
> > >
> > > ---
> > >
> > > We hope these additional results and clarifications help resolve the remaining concerns.

---

### Decision · Program_Chairs · 2026-04-30

**Decision:**

Accept (regular)

**Comment:**

This paper focuses on the problem of multi-view 3D human pose estimation and models the problem of associating individuals from multiple views as generative process. The paper received three Weak Accept and one Accept rating. The reviewers initially raised some concerns, specifically related to scalability, robustness, reliance on pseudo labels, the limited analysis of the failure cases and the use of a relatively small benchmark. However, the replies of the authors seem to have eliminated the majority of the issues of the reviewers, who believe this is a technically solid paper. Given the unanimous accept ratings, the AC finds no reason to overturn the reviewers' decision. With that being said, the AC would encourage the authors to consider all comments of the reviewers when preparing the final version of the paper.